# Cell-mediated exon skipping normalizes dystrophin expression and muscle function in a new mouse model of Duchenne Muscular Dystrophy

Francesco Galli [1]✉, Laricia Bragg[1], Maira Rossi [1], Daisy Proietti[2], Laura Perani [2], Marco Bacigaluppi [2], Rossana Tonlorenzi [2], Tendai Sibanda[1], Miriam Caffarini[1], Avraneel Talapatra [1], Sabrina Santoleri[1], Mirella Meregalli[3], Beatriz Bano-Otalora [4], Anne Bigot [5], Irene Bozzoni [6,7], Chiara Bonini[8,9], Vincent Mouly[5], Yvan Torrente [3] & Giulio Cossu [1,2,10]✉

## Abstract

Cell therapy for muscular dystrophy has met with limited success, mainly due to the poor engraftment of donor cells, especially in fibrotic muscle at an advanced stage of the disease. We developed a cell-mediated exon skipping that exploits the multinucleated nature of myofibers to achieve cross-correction of resident, dystrophic nuclei by the U7 small nuclear RNA engineered to skip exon 51 of the dystrophin gene. We observed that co-culture of genetically corrected human DMD myogenic cells (but not of WT cells) with their dystrophic counterparts at a ratio of either 1:10 or 1:30 leads to dystrophin production at a level several folds higher than what predicted by simple dilution. This is due to diffusion of U7 snRNA to neighbouring dystrophic resident nuclei. When transplanted into NSG-mdx-Δ51mice carrying a mutation of exon 51, genetically corrected human myogenic cells produce dystrophin at much higher level than WT cells, well in the therapeutic range, and lead to force recovery even with an engraftment of only 3–5%. This level of dystrophin production is an important step towards clinical efficacy for cell therapy.

**Keywords** Duchenne Muscular Dystrophy; Cell Therapy; Exon Skipping; Mesoangioblast; Regenerative Medicine
**Subject Categories** Genetics, Gene Therapy & Genetic Disease; Musculoskeletal System; Stem Cells & Regenerative Medicine

## Introduction

Duchenne Muscular Dystrophy (DMD) is the most common and one of the most severe muscular dystrophies, affecting approximately one in four thousand newly born children (Emery, 2002). It is characterized by progressive wasting of skeletal and cardiac muscle, leading to a variable but progressive muscle weakness that limits the patient's motility and, in later years affects cardiac and respiratory functions (Muntoni et al, 2003). Duchenne Muscular Dystrophy (DMD) is caused by different mutations of the dystrophin gene, located on the X chromosome (Worton et al, 1984; Nallamilli et al, 2014). In 90% of cases, mutations lead to a change in the mRNA reading frame that prevents dystrophin protein production (Bladen et al, 2015). In frame deletions lead to a shorter but partially functional dystrophin, associated with milder Becker Muscular Dystrophy (BMD) (Den Dunnen et al, 1989).

Dystrophin and the dystrophin-associated glycoproteins (e.g., sarcoglycans) have a critical role in muscle cell interaction with the basal lamina and provide elastic resistance to the sarcolemma during contraction. In the absence of dystrophin and associated proteins, the membrane is more easily damaged, leading to calcium influx, hyper contraction, proteolysis and fibre degeneration (Davies and Nowak, 2006). Degeneration is followed by regeneration carried out by satellite cells, resident myogenic stem/progenitor cells and, to a minor extent, interstitial cells such as pericytes (Biressi et al, 2020). In humans, adult myogenic cells have limited self-renewal potency and in DMD the continuous regeneration cycles eventually lead to depletion of the myogenic cell populations. Muscle degeneration is accompanied by chronic inflammation that progressively leads to accumulation of dense connective and adipose tissues that replace muscle fibres (Klingler et al, 2012) making any therapy ineffective at this stage.

[1]Division of Cell Matrix Biology & Regenerative Medicine, Faculty of Biology, Medicine and Health, University of Manchester, Manchester, UK. [2]Institute of Experimental Neurology, Division of Neurosciences, IRCCS San Raffaele Scientific Institute, Milan, Italy. [3]Department of Pathophysiology and Transplantation, Università degli Studi di Milano, Fondazione IRCCS Ca' Granda Ospedale Maggiore Policlinico, Centro Dino Ferrari, 20122 Milan, Italy. [4]Division of Neuroscience and Experimental Psychology, School of Biological Sciences, Faculty of Biology, Medicine and Health, University of Manchester, Manchester, UK. [5]Institut de Myologie, Université Pierre et Marie Curie, Paris 6 UM76, Univ. Paris 6/U974, UMR7215, CNRS, Pitié-Salpétrière-INSERM, UMRS 974 Paris, France. [6]Department of Biology and Biotechnology Charles Darwin, Sapienza University of Rome, 00161 Rome, Italy. [7]Center for Life Nano- & Neuro-Science@Sapienza of Istituto Italiano di Tecnologia (IIT), 00161 Rome, Italy. [8]Experimental Hematology Unit, Vita-Salute San Raffaele University, Milan, Italy. [9]IRCCS Ospedale San Raffaele Scientific Institute, 20133 Milan, Italy. [10]Experimental and Clinical Research Center. Charité Medical Faculty and Max Delbrück Center 13125 Berlin, Berlin, Germany. ✉E-mail: francessco.galli@manchester.ac.uk; giulio.cossu@manchester.ac.uk

## The paper explained

### Problem

Duchenne Muscular Dystrophy (DMD) is a genetic disease affecting primarily skeletal and cardiac muscle, progressively limiting patient motility, and resulting in respiratory and cardiac failure which are the cause of premature death. All new therapeutic strategies have shown very modest or no efficacy in clinical trials. One common problem is the delivery of the therapeutic agent (virus, cell or macromolecule) to skeletal muscles, the most abundant tissue of the human body, which results in little colonization/local concentration and thus small or no therapeutic effect.

### Results

We devised a new strategy combining cell therapy and exon skipping. Cells from the patient have been corrected in culture with a lentiviral vector expressing a small nuclear RNA that cause skipping of exon 51 of the dystrophin gene, the most common mutation that causes DMD. Corrected cells fuse with resident myogenic cells into a regenerating muscle fibre, both in culture and after injection in immune deficient DMD mice. The genetically corrected cell produces a small nuclear RNA that is able to enter not only the nucleus of origin but also into all the neighbouring nuclei of the muscle fibre, amplifying the amount of dystrophin to a level considered therapeutic and improving the motility of transplanted mice to the level of normal mice.

### Impact

Our findings show that by combining cell transplantation and exon skipping strategy, it is possible to obtain a synergistic effect and a reduction of their relative drawbacks. Based on these results a phase I/IIa clinical trial on DMD patients is starting. Moreover, the same strategy may be applied to other mutations of the dystrophin gene and to other muscular dystrophies where the mutation affects a large gene (e.g., dysferlin for Limb Girdle Muscular Dystrophy 2B) whose cDNA cannot be accommodated into currently available viral vectors.

Thirty-six years since the cloning of the dystrophin gene (Hoffman et al, 1987) there is still no cure for DMD, although in the last two decades significant progress has been made in its clinical management. Many novel therapeutic approaches have entered the clinical arena, but none has yet reached significant and long-lasting clinical efficacy. These approaches include exon skipping, gene therapy, PTC124 and cell-based therapy (Galli et al, 2018). Oligonucleotide-based exon skipping therapy failed to reach efficacy in a phase III trial possibly because of the inability of these large molecules to diffuse through specific barriers such as collagen-rich fibrotic tissue and muscle basal lamina (Godfrey et al, 2017). Current focus converges on AAV-mediated in vivo gene therapy to deliver either mini-dystrophin or nucleases to repair the mutation in situ (Duan, 2018; Hotta, 2015). While this would be the straightest solution, serious problems persist: unknown long-term persistence, immunogenicity including pre-existing immunity, costs of production and, most important, toxicity associated to the high dose needed for efficient expression. Toxicity has been noticed in virtually any trial and culminated with the death of three patients (https://musculardystrophynews.com/2021/12/23/pfizer-reports-death-duchenne-md-gene-therapy-trial-participant/) and (https://www.cgtlive.com/view/dmd-patient-dies-in-crispr-gene-therapy-trial-led-by-nonprofit-biotech-cure-rare-disease/) (Lek et al, 2023). In contrast, cell therapy, even if technically more demanding, promises a permanent effect, since muscle fibres last in

principle for a lifetime, and this would avoid both life-long drug administration and immune suppression, in the case of autologous, genetically corrected cells. After ten years of preclinical work both in small and large animal models (Sampaolesi et al 2003, 2006; Tedesco et al 2011, 2012) a "first in man" Phase I/IIa cell therapy trial of DMD was completed using mesoangioblasts, pericyte-derived mesoderm progenitor cells, from an HLA-matched sibling. Results showed safety but very low efficacy (Cossu et al, 2015), due to the advanced age of the patients (9–14 years at the time of the recruitment), an administered dose lower than in pre-clinical work, and, most importantly, a low engraftment (<1%) of donor cells in dystrophic muscle. Even starting at an earlier age and implementing the protocol, engraftment will always remain low in tissues such as muscle and brain where ablation of diseased resident cells is not possible (Cossu et al, 2018). Consequently, the transplanted healthy cells, once fused into a regenerated muscle fibre, need to produce enough dystrophin to compensate for all resident nuclei that cannot do so.

In this paper we develop a new strategy combining the advantage of exon skipping with that of cell-based therapy. In order to test our hypothesis we used either Telomerase-Immortalised Myogenic cells (TIM) (Mamchaoui et al, 2011), MyoD-converted fibroblasts (Lattanzi et al, 1998) or primary human Mesoangioblasts (hMabs) from a DMD patient, all carrying a frameshift mutation in exon 51. Cell-mediated exon-skipping had already been shown to cause dystrophin synthesis in human CD133+ cells, and these cells were able to express dystrophin in culture and to colonise and produce human dystrophin in immune-deficient mdx mice though at a low level (Benchaouir et al, 2007). This was because the mdx mouse harbours a different mutation, in exon 23, that cannot be skipped by the U7 snRNA designed to induce skipping of human exon 51. Therefore, we generated a new mouse model that harbours a skippable mutation of a humanized exon 51. Myogenic cells were transduced with a lentiviral vector expressing the U7 snRNA, able to induce skipping of human exon 51, that we first described 20 years ago (De Angelis et al, 2002); this snRNA diffuses along the myofiber and permanently corrects also dystrophic resident nuclei in the same area. We show that this strategy multiplies the therapeutic effect both in vitro and in newly regenerated muscle fibres in the in vivo DMD model, leading to abundant dystrophin production, in a range that is considered therapeutically effective (Kinali et al, 2008).

## Results

### Cell-mediated exon skipping induced by lentiviral vector U7#51T2AGFP

In order to work with a long term expandable myogenic cell line, and to avoid inter-patient variability, we choose a line of immortalised myogenic cells carrying a deletion at exon 48–50 (Δ48–50). The cells (defined TIM) were immortalised with telomerase and CDK4, can be indefinitely expanded and even at late passages maintain reproducible and robust myogenic differentiation (Mamchaoui et al, 2011). We also used a similarly immortalised WT type control line, with similarly high differentiation potency (Mamchaoui et al, 2011). We confirmed results also in primary mesoangioblasts from a DMD patient and in DMD

fibroblasts myogenically converted with an Adenovector expressing MyoD (Lattanzi et al, 1998), all with the same mutation.

To correct the genetic defect via exon skipping, we built a 3° generation, SIN lentiviral vector (LV) expressing, under the transcriptional control of the human E1F-α ubiquitous promoter, the U7 small nuclear RNA (De Angelis et al, 2002), engineered to skip the acceptor and donor splice sites of exon 51, thus causing restoration of the correct reading frame. To facilitate detection of transduced cells the vector also expresses GFP after a T2A site (Fig. EV1). A control vector, only expressing GFP was also produced.

Co-culture of DMD transduced and non-transduced cells at a ratio of 1:10 and 1:30 led to a progressive reduction of fluorescence (Fig. EV2C).

We tested the ability of the U7#51T2AGFP LV to induce cell-mediated exon skipping and thus a shorter version of dystrophin. Total RNA was extracted from DMD differentiated myotubes, either corrected or not with the LV and the generated cDNA was used to perform a RT-PCR. A human muscle biopsy was utilised as a positive control to show the size of the full-length Dystrophin (900 bp). A band of approximately 400 bp was observed in all samples from DMD TIM myotubes. A second dystrophin band of approximately 250 bp (size of the shorter Dystrophin) was visible only in the cells transduced with the LV showing the dystrophin mRNA produced through exon-skipping by the U7#51T2AGFP (Fig. 1A). Sequencing of the 250 bp band showed that it corresponds to Δ47−52 (Fig. 1B). This result confirms the ability of U7#51T2AGFP to induce exon-skipping. To test the ability of Δ47−52 dystrophin mRNA to produce a protein, immunofluorescence (IF) staining was carried out in DMD TIM cells and genetically corrected DMD TIM cells (herein referred to as DMD-U7). Moreover, we hypothesized that the U7 snRNA would diffuse to neighbouring DMD nuclei and correct also the dystrophin produced in these nuclei (Fig. EV2A).

By taking advantage of GFP expression in both vectors, we quantified their ability to transduce myogenic cells. Figures 1C and EV2C show that at a low MOI (0.8) approximately 80% of cells were transduced by the U7#51T2AGFP LV and more than 90% by the control GFP LV, as revealed by FACS analysis (Fig. 1C). Therefore, DMD-U7 TIM cells were co-cultured with non-corrected DMD TIM cells at a 1:10 and 1:30 ratio, mimicking an in vitro cell engraftment of 10% and 3% respectively, the second in the same order of magnitude of the 0.7% engraftment observed in the youngest patient of the previous clinical trial (Fig. EV2C) (Cossu et al, 2015). Figure 1D shows merged images of the 1:10 and 1:30 co-cultures where the expression of Dystrophin (red) extends well beyond the areas expressing GFP (green) in the cytoplasm of multinucleated myotubes as indicated by arrows. Individual fluorescence channels are shown in Fig. EV2A and EV2B. This observation would support our hypothesis of a possible cross correction by U7#51T2AGFP due to its diffusion in the neighbouring dystrophic nuclei. We next tested the possibility that snRNA may spread and induce exon skipping in the neighbouring dystrophic nuclei by qRT-PCR on DMD-U7 TIM and DMD TIM cells co-cultured at a 1:10 and 1:30 ratio (Fig. 1E). MyHC expression was used to quantify the level of the differentiation and no significant difference was noted across all the conditions (Fig. 1E), confirming the IF analysis (Fig. EV2A and EV2B). GFP expression was directly compared with the dystrophin expression to evaluate the level of dystrophin produced by cross-correction.

GFP expression decreased proportionally to the ratio of transduced vs non transduced cells. In contrast, the 1:10 and 1:30 co-cultures showed higher expression of dystrophin in comparison to GFP. We observed a dystrophin expression of almost 40% (fourfold increase) and 20%, in comparison with WT cells, (about sevenfold increase) in cells co-cultured at 1:10 ratio and 1:30 ratio, respectively, (Fig. 1E), supporting the possibility that the snRNA produced by U7#51T2AGFP may induce cross correction of neighbouring nuclei, thus amplifying dystrophin production.

## Intra-cellular diffusion of snRNA produced by U7#51T2AGFP

To demonstrate the possible intra-cytoplasmic diffusion of the U7 snRNA from the original transduced nucleus into neighbouring dystrophic nuclei of the same myotube, we conducted double in situ hybridization with specific mRNA probes. One probe was directed against the U7 snRNA (Red) and the other against GFP mRNA (Green). Whereas the GFP mRNA was localised in the cytoplasm surrounding a single, donor nucleus (green arrows), the snRNA diffused along most of the available cytoplasm in the same myotube (red arrows) also were no GFP mRNA expressing nuclei could be detected (Fig. 2A). These results demonstrate the ability of the U7#51T2AGFP snRNA to diffuse along the myotube and to enter also in the neighbouring nuclei which had not been genetically corrected.

## Dystrophin protein expression induced by U7 snRNA in vitro

To quantify the protein expression produced by nuclear cross correction we performed a Western Blot assay (WB).

Proteins were extracted from co-cultures of DMD-U7 TIM and DMD TIM cells at 1:10 and 1:30 ratio. As a control, WT TIM cells were co-cultured with DMD TIM cells under the same conditions. The level of differentiation was quantified by analysing Myosin Heavy Chain (MyHC) that showed no significant difference in expression among the different samples (Fig. 2B). As expected, DMD TIM cells did not produce any dystrophin protein while WT TIM cells robustly did so. TIM DMD cells after transduction (DMD-U7 TIM) also expressed dystrophin, to a lower level than WT TIM cells. However, at 1:10 dilution, dystrophin expression was already higher in co-cultures of DMD TIM with DMD-U7 TIM cells than in co-cultures of DMD TIM with WT TIM cells. At 1:30 dilution, dystrophin was still clearly detectable in co-cultures of DMD TIM and DMD-U7 TIM cells but not in co-cultures with WT TIM cells. The quantification of the WB (densitometry) showed a robust dystrophin expression in DMD TIM cells co-cultured with DMD-U7 cells at 1:10 and 1:30 ratio (50% and 20% of WT TIM cells) in comparison with the co-culture WT TIM cells and DMD TIM cells at the same ratio, showing only a small increase (15% at 1:10 ratio and undetectable at 1:30 ratio). To demonstrate that exon skipping cross-correction via snRNA is effective also with other types of myogenic cells, we co-cultured DMD-U7 TIM and WT TIM cells with two different DMD myogenic cell types, either mesoangioblasts (DMDhMABs) or MyoD-converted Fibroblasts (DMDMyoD^ERTFbs) derived from DMD patients with the same mutation. Dystrophin protein expression was detected in all the different co-cultured samples (Fig. 2C,D), always with the same

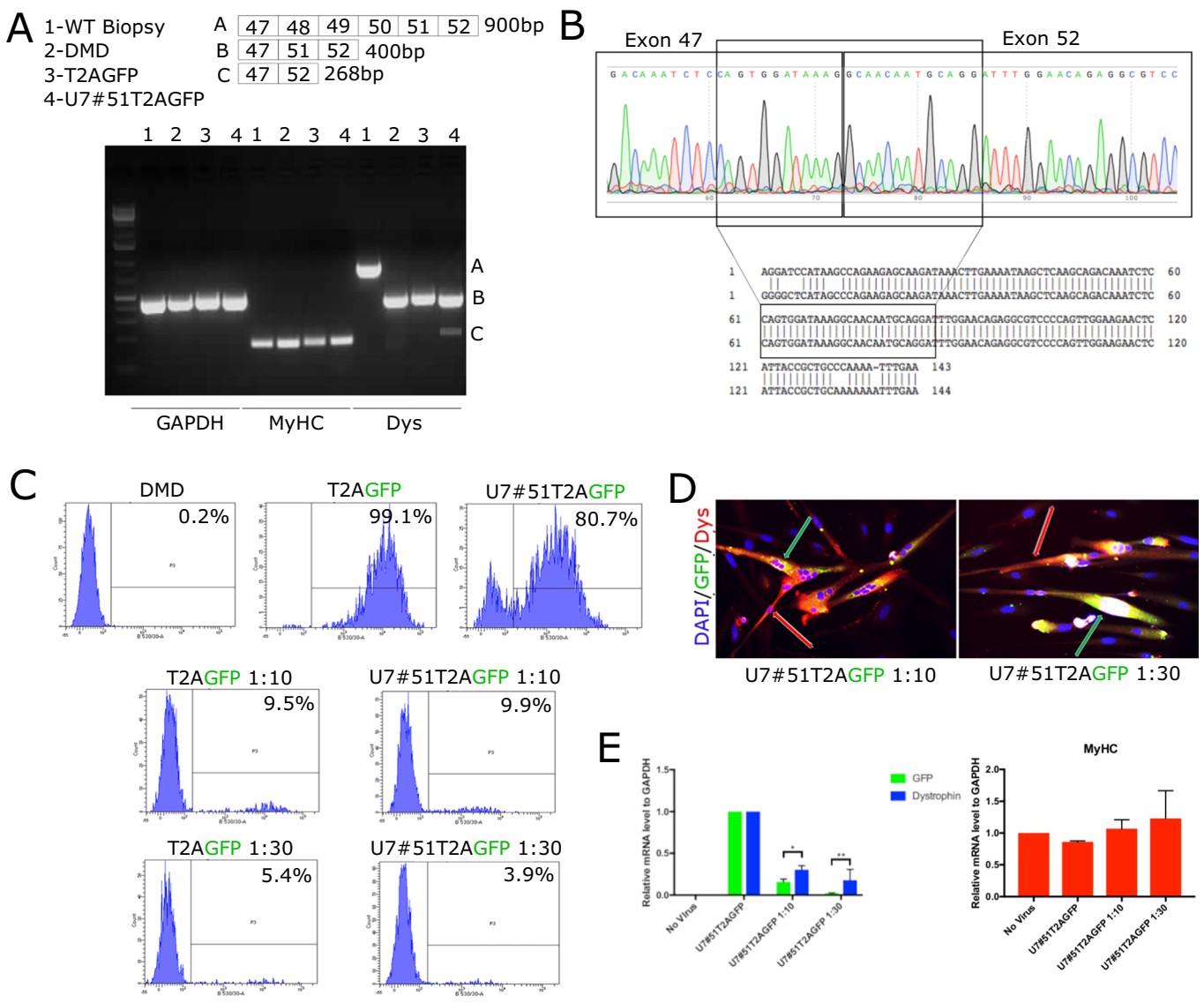

**Figure 1. Cell-mediated exon skipping induced by lentiviral vector U7#51T2AGFP.**

(A) Representative RT-PCR images of dystrophin transcripts from indicated sample ($n = 3$ biological replicates). (B) Representative sequencing of the cDNA from the band in (A), showing the expected sequence for exon 47–52. (C) Gating strategy used to quantify the percentage of DMD TIM cells transduced with the vectors expressing either GFP (T2AGFP) or U7 snRNA (U7#51T2AGFP) in a mixture (1:10 or 1:30) of transduced and non-transduced cells ($n = 3$ biological replicates). (D) Representative IF images showing expression of Dystrophin and GFP in a DMD-U7/DMD TIM cells co-culture ($n = 3$ biological replicates). (E) qRT-PCR analysis of dystrophin and GFP mRNA level in a DMD-U7/DMD TIM cells co-culture ($n = 3$ biological replicates). Data Information: Data are shown as mean ± SD. *$P < 0.05$, **$P < 0.01$. Multiple *t*-test are followed by Bonferroni correction unpaired *t*-test. The mean differences between two group were calculated with two-way anova. Experiments have been replicated for at least three times. Source data are available online for this figure.

trend. Quantification of protein expression indicated a higher dystrophin expression in co-cultures of DMD cells with DMD-U7 TIM cells in comparison with co-cultures of DMD cells with the WT TIM cells. This result confirms that the phenomenon is not specific of immortalised cells, and in all conditions dystrophin protein is likely produced also in neighbouring dystrophic nuclei corrected by the U7 snRNA produced by genetically corrected nuclei.

These data demonstrated that our snRNA-based strategy enhances the production of dystrophin protein through cross-

correction of dystrophic neighbouring nuclei within the same myotube.

## Dystrophin protein expression induced by U7 snRNA after intra-muscular injection in vivo

Currently used mdx mice carry a mutation in exon 23 that is not amenable to cell-mediated exon skipping by a snRNA engineered to skip human exon 51. For this reason, we generated at Jackson Labs a NSG mouse carrying a skippable mutation of exon 51. The mice

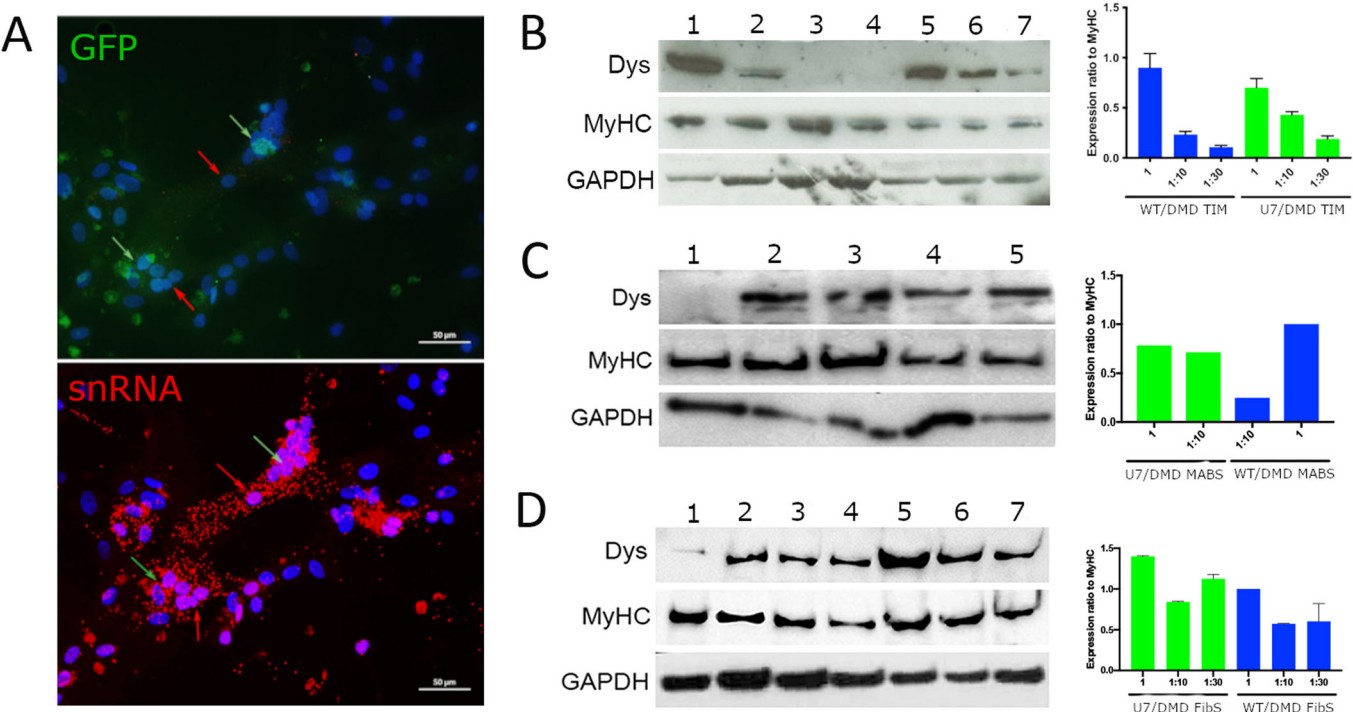

**Figure 2. Intracellular diffusion of snRNA and Dystrophin protein expression.**

(A) Representative in situ hybridization images showing the localization of GFP mRNA and of U7 snRNA in myotubes of a DMD-U7/DMD TIM cells co-culture ($n = 3$ biological replicates). (B) Representative WB image of Dystrophin expression from differentiated cultures of TIM cells. Samples are from left to right: 1: WT TIM cells; 2: WT TIM cells co-cultured with a 10-fold excess of DMD TIM cells; 3: WT TIM cells co-cultured with a 30-fold excess of DMD TIM cells; 4: DMD TIM cells; 5: U7#51T2AGFP transduced DMD TIM cells; 6: U7#51T2AGFP transduced DMD TIM cells co-cultured with a 10-fold excess of DMD TIM cells; 7: U7#51T2AGFP transduced DMD TIM cells co-cultured with a 30 fold excess of DMD TIM cells. On the right quantification of Dystrophin expression ($n = 3$ biological replicates). (C) Representative WB image of Dystrophin expression from differentiated cultures of hMABs. Samples are from left to right: 1: DMD hMABs; 2: DMD-U7 cells; 3: DMD-U7 cells co-cultured with a 10-fold excess of DMD hMABs; 4: WT TIM cells co-cultured with a 10-fold excess of DMD hMABs; 5: WT hMABs cells. On the right quantification of Dystophin expression ($n = 3$ biological replicates). (D) Representative WB image of Dystrophin expression from differentiated cultures of DMD MyoD-converted Fbs. Samples are from left to right: 1: DMD MyoD-converted Fbs; 2: DMD-U7 cells; 3: DMD-U7 cells co-cultured with a 10 fold excess of DMD Myod-converted Fbs; 4:DMD-U7 cells co-cultured with a 30 fold excess of DMD Myod-converted Fbs 5: WT TIM cells; 6: WT TIM cells co-cultured with a 10 fold excess of DMD Myod-converted Fbs; 7: WT TIM cells co-cultured with a 30 fold excess of DMD Myod-converted Fbs. On the right quantification of Dystrophin expression ($n = 3$ biological replicates). Data Information: Data are shown as mean ± SD. Multiple $t$-test are followed by Bonferroni correction unpaired $t$-test. Experiments have been replicated for at least three times. Source data are available online for this figure.

(defined for simplicity NSG-mdx-Δ51) are completely immune deficient and develop a muscular dystrophy like the classic mdx, with a peak of degeneration about the first month of age and a progressive but relatively mild dystrophic phenotype that worsens after one year of age (Fig. EV3A). Therefore, when human cells are transplanted into these mice, they will fuse with resident dystrophic myogenic cells to form hybrid regenerating myofibers. Importantly, mouse nuclei could be corrected since also acceptor and donor splice sites of exon 51 had been humanised.

In a preliminary experiment, carried out in 6 months old mdx/SCID mice ($n = 3$), we repeated the in vitro dilution experiments by co-culturing DMD-U7 TIM cells with DMD TIM cells at 1:10 ratio in a Matrigel plug implanted under the dorsal skin of 100% DMD-U7 TIM cells were used as control in a similar Matrigel plug implanted under the dorsal skin on the contra-lateral site of the same mouse (Fig. EV4A). After a month, we recovered plugs containing multinucleated GFP+ myofibers only in two mice. WB analysis showed virtually the same amount of dystrophin protein detected both in the plug containing only DMD-U7 TIM cells and

in those containing 1/10 of DMD-U7 TIM and 9/10 of DMD TIM cells (Fig. EV4B).

When NSG-mdx-Δ51mice became available, 6 animals, per experimental group, were injected at 6 months of age with the same number ($5 \times 10^5$ cells) of either DMD-U7 TIM or WT TIM cells. In other 6 animals we also injected, as controls, either DMD-U7 hMABs or WT hMABs. To evaluate Dystrophin correction at the protein level we performed WB analysis of the injected *Tibialis Anterior* (TA), 1 month after a single intramuscular injection. WB analysis showed restoration of dystrophin expression, both with TIM cells and hMABs (Fig. 3A,B), in the skeletal muscle. As shown in Fig. 3A the TA injected with DMD-U7 TIM cells showed approximately 60% of protein level detected in WT muscle while the TA injected with WT TIM cells showed protein restoration but a lower-level in comparison with DMD-U7 cells (approximately 20%) suggesting the ability of the snRNA to induce expression also in the neighbouring nuclei which had not genetically corrected. A similar result was observed in the TA injected with DMD-U7 hMABs and WT hMABs (Fig. 3B) respectively with around 50%

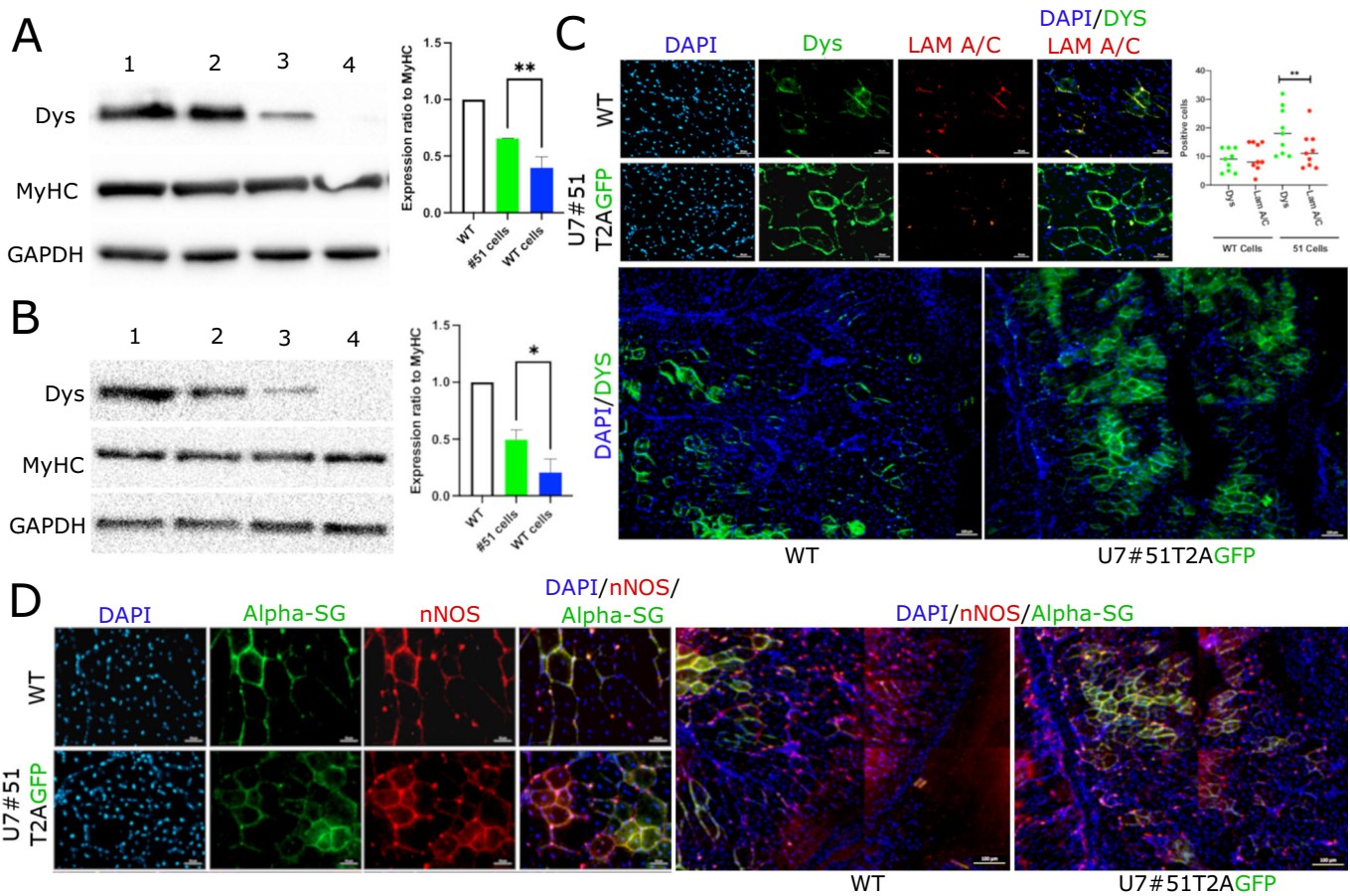

**Figure 3. In vivo Dystrophin protein expression induced by lentiviral vector U7#51T2AGFP.**

(A) Representative WB image of Dystrophin expression in a TA of NSG-mdx-Δ51or NSG WT mice. Samples are from left to right: 1: NSG WT muscle; 2: NSG-mdx-Δ51muscle transplanted with $5 \times 10^5$ DMD-U7 cells; 3: NSG-mdx-Δ51muscle transplanted with $5 \times 10^5$ WT TIM cells; 4: NSG-mdx-Δ51 muscle non transplanted. On the right quantification of Dystrophin expression ($n = 6$ biological replicates). (B) Representative WB image of Dystrophin expression in a TA of NSG-mdx-Δ51or NSG WT mice. Samples are from left to right: 1: NSG WT muscle; 2: NSG-mdx-Δ51muscle transplanted with $5 \times 10^5$ U7 hMABs; 3: NSG-mdx-Δ51 muscle transplanted with $5 \times 10^5$ WT hMABs; 4: NSG-mdx-Δ51 muscle non transplanted. On the right quantification of Dystrophin expression ($n = 6$ biological replicates). (C) Representative IF images showing expression of Dystrophin and Lamin AC of transverse sections of TA of NSG-mdx-Δ51 mice transplanted either with $5 \times 10^5$ WT TIM cells or with $5 \times 10^5$ DMD-U7 cells. In the inset, Quantification of dystrophin positive fibres ($n = 6$ biological replicates). (D) Representative IF images showing expression of Alpha-SG and nNOS of transverse sections of TA of NSG-mdx-Δ51 mice transplanted either with $5 \times 10^5$ DMD-U7 cells or with $5 \times 10^5$ WT TIM cells ($n = 6$ biological replicates). Data Information: Data are shown as mean ± SD. *$P < 0.05$, **$P < 0.01$. Multiple $t$-test are followed by Bonferroni correction unpaired $t$-test. The mean differences between two group were calculated with two-way anova. Experiments have been replicated for at least three times. Source data are available online for this figure.

and 10% of dystrophin expression. 30% of normal level of dystrophin is estimated to provide therapeutic benefits for DMD patients (Godfrey et al, 2015). IF of muscle sections from injected TA with either WT TIM or DMD-U7 cells confirmed recovery of dystrophin (Fig. 3C) α-sarcoglycan and nNOS (Fig. 3D) expression.

To test whether the very high level of dystrophin expression was due to the genetic correction of resident nuclei, mediated by intra-cytoplasmic diffusion of the U7 snRNA from the nucleus of the genetically corrected DMD cells that had been transplanted, we conducted double in situ hybridization with specific mRNA probes, one directed against the snRNA (Red) and the other against human LAM A/C (Green). The snRNA diffused along most of the available cytoplasm in the same myofiber also were no LAM A/C mRNA expressing nuclei could be detected (Fig. 4A). These results demonstrate the ability of the U7#51T2AGFP snRNA to diffuse along the myofiber and to enter the neighbouring nuclei which had

not been genetically corrected, thus explaining the efficacy of a single cell injection in restoring the protein expression of dystrophin, up to around 60% of WT levels, a result only achievable so far with injection of AAV.

We reasoned that lowering the dose of cells injected, should lead to a less than proportional decrease in the amount of dystrophin produced, because of the mechanism discussed above. To test this hypothesis, we injected in the TA of 6 NSG-mdx-Δ51mice 6 months old, 5, 2.5 and 1.25 and $0.62 \times 10^5$ DMD-U7 TIM cells. After one month we performed a WB analysis of the extracted proteins. Figure 4B shows that even at the lowest cell dose we could detect about 50% of the maximum dose injected (rather than the expected 12.5%) as normalised by the analysis of LAM A/C that detects the number of human cells present. This result is relevant to the idea of future studies in patients where the actual dose of cells administered will necessary be lower based on the large number of muscles to be treated.

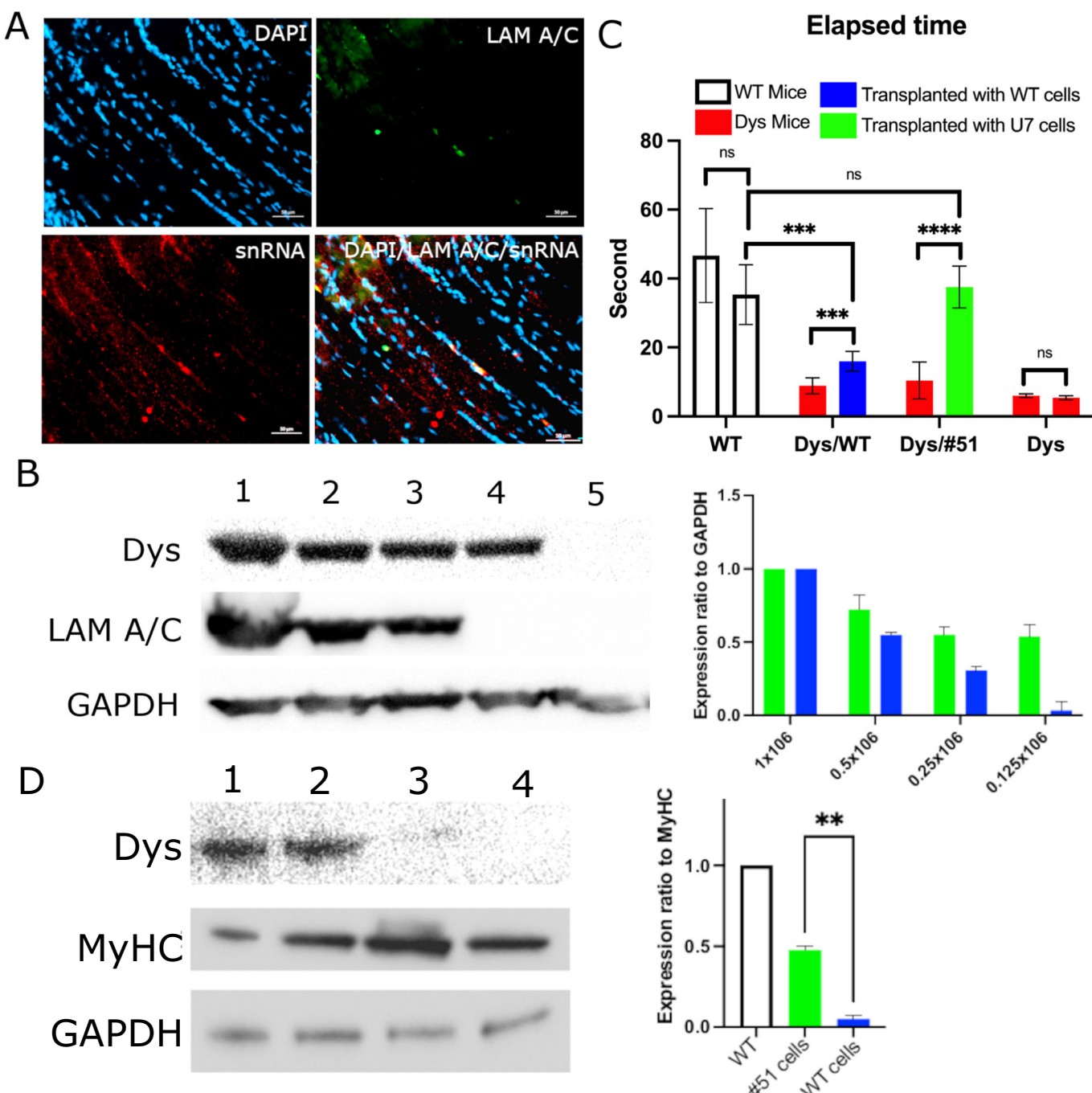

**Figure 4. Intracellular diffusion of snRNA *in vivo* and improving of motility after transplantation.**

(A) Representative in situ hybridization images showing the localization of Lamin AC and of U7 snRNA in a transverse section of TA of NSG-mdx-Δ51 mice transplanted with $5 \times 10^5$ DMD-U7 cells (*n* = 6 biological replicates). (B) Representative WB image of Dys expression in a TA of NSG-mdx-Δ51: samples are from left to right: 1: NSG-mdx-Δ51muscle transplanted with $5 \times 10^5$ DMD-U7 cells; 2: NSG-mdx-Δ51muscle transplanted with $2.5 \times 10^5$ DMD-U7 cells; 3: NSG-mdx-Δ51 muscle transplanted with $1.25 \times 10^5$ DMD-U7 cells. 4: NSG-mdx-Δ51 muscle transplanted with $0.625 \times 10^5$ DMD-U7 cells.5: NSG-mdx-Δ51 muscle not transplanted. On the right quantification of Dys expression (*n* = 6 biological replicates). (C) Treadmill motility assay analysis before and after transplantation with either $5 \times 10^5$ WT TIM or $5 \times 10^5$ DMD-U7 cells (*n* = 6 biological replicates). (D) Representative WB image of Dys expression in a TA of NSG-mdx-Δ51 one year after the transplant: samples are from left to right: NSG-WT muscle; NSG-mdx-Δ51muscle transplanted with $5 \times 10^5$ DMD-U7 cells; 2: NSG-mdx-Δ51muscle transplanted with $5 \times 10^5$ WT TIM cells; NSG-mdx-Δ51muscle not transplanted. On the right quantification of Dys expression (*n* = 6 biological replicates). Data Information: Data are shown as mean ± SD. \**P* < 0.05, \*\**P* < 0.01, \*\*\**P* < 0.001, \*\*\*\**P* < 0.0001. Multiple *t*-test are followed by Bonferroni correction unpaired *t*-test. The mean differences between two group were calculated with two-way anova. Experiments have been replicated for at least three times. Source data are available online for this figure.

An important issue that needs to be addressed is the duration of the expected therapeutic effect: will a single injection of WT TIM or DMD-U7 TIM cells allow long term survival of the transplanted fibres and prevent their subsequent degeneration thanks to the dystrophin produced? To test this, we injected the same number of the two cell types in the *Tibialis Anterior* (TA) of NSG-mdx-Δ51mice, 6 month old, and after 11 months collected the muscles and analysed the amount of dystrophin by WB Fig. 4D shows that after almost one year the amount of dystrophin in the TA transplanted with DMD-U7 TIM, is comparable with the level detected after one month (approximately 50% of a WT muscle) while dystrophin is barely detected in the TA of mice transplanted with WT TIM. This suggests that the amount of dystrophin produced also by trans-corrected resident dystrophic nuclei is sufficient to support long term survival of transduced fibres, something that may not happen when the amount of dystrophin produced by the few WT nuclei may not be sufficient to maintain long term integrity of the fibre.

We also evaluated whether the increased production of dystrophin would ameliorate the animal motility. Both *Tibialis Anterior* (TA), the *Gastrocnemius* (Gas), the *Quadriceps* (Q) were transplanted with either WT TIM or DMD-U7 TIM cells ($5 \times 10^5$ cells per muscle) in 6-months-old animal. After 21 days we challenged the animals, for two consecutive days, on a treadmill recording the distance and running time. Before transplant NSG-mdx-Δ51mice showed a 70% reduction of motility when compared with NSG WT mice (Fig. 4C). Mice after transplant with DMD-U7 cells, showed a recovery of motility reaching almost the same level of WT animals, while mice transplanted with WT TIM cells showed an improve in their motility, but not a full recover, in comparison with WT animals (Figs. 4C and EV5).

### Dystrophin protein expression induced by U7 snRNA after intra-arterial injection in vivo

Finally, we tested the ability of DMD-U7 hMABs to migrate from the femoral artery to the downstream muscles, fuse with regenerating muscle fibres and to induce the exon skipping also in neighbouring nuclei. To this aim, we performed a single intra-arterial injection of $5 \times 10^5$ cells, either DMD-U7 hMABs or WT hMAbs in 6 months old, NSG-mdx-Δ51mice (5 mice per group). One month after the injection, WB analysis of the *Gastrocnemius* muscle showed that DMD-U7 hMABs produced approximately 10% of normal level of dystrophin while WT hMABs did not produce an amount of dystrophin detectable by WB (Fig. 5A). Of note, in previous work, we needed multiple consecutive injections to detect the WT copy of the mutated gene product by WB (Sampaolesi et al, 2003, 2006). As shown in Fig. 5B, IF of muscle sections of the *Tibialis Anterior* intra-arterially injected with WT hMABs showed some dystrophin positive fibres containing human nuclei, identified by LAM A/C antibodies; however, the *Tibialis Anterior* intra-arterially injected with DMD-U7 hMABs showed many more dystrophin positive fibres only some of which contained human nuclei, confirming expression also in neighbouring dystrophic mouse nuclei (Fig. 5C).

## Discussion

Thirty-six years after the cloning of the dystrophin gene (Hoffman et al, 1987), Duchenne Muscular Dystrophy still lacks an efficacious

therapy. Hopes for a successful treatment followed gene identification, myoblast transplantation (Partridge et al, 1989), adenoviral vector-mediated gene therapy (Ragot et al, 1993), oligo-mediated exon skipping (Kinali et al, 2009), PTC 124-mediated skipping of premature termination code (Finkel et al, 2013), mesoangioblast-mediated cell therapy (Cossu et al, 2015) and more recently either AAV-mediated expression of micro-dystrophin (Duan, 2018) or CRISPR-Cas9 for genome correction (Tabebordbar et al, 2016). All the previous attempts essentially failed, even when they reached phase III trials and marketing authorization; AAV-mediated gene therapy is currently considered the most promising strategy, but problems related to immunity, severe toxicity (including the recent death of three patients), duration of treatment and costs all remain to be solved (Arnold, 2021).

We have been pursuing for over twenty years a cell therapy approach, but the first clinical trial showed minimal efficacy, related to a very low engraftment of donor mesoangioblasts. Since engraftment can be ameliorated only marginally in tissues when resident cell ablation is not possible (Cossu et al, 2018), we developed an alternative strategy that combines cell therapy and exon skipping. In other words, we used donor cells as trojan horses to fuse with regenerating muscle fibres and produce the U7 small nuclear RNA that enters also neighbouring resident nuclei, thus amplifying several folds dystrophin production and hence the presumed therapeutic effect.

Importantly, we verified that the U7 snRNA diffuses from the transplanted nucleus along the cytoplasm of the muscle fibre, in vitro but also in vivo, even though the cytoplasm is mostly occupied by the contractile apparatus. However, muscle nuclei are aligned in chains (located in the centre of regenerating fibres or under the sarcolemma in mature fibres) and there are no sarcomeres in between them, thus creating a space where the snRNA may freely diffuse.

We also found that decreasing numbers of transplanted cells maintain an expression of dystrophin well above the level expected by simple dilution. Moreover, a single injection was sufficient to maintain dystrophin expression for 11 months, a significant period of a mouse lifespan (Fig. 4D); even if expression may decrease with time, due to persistent degeneration/regeneration cycles of skeletal muscle, we should consider that, at variance with AAV, autologous cells can be administered repeated times (Cossu et al, 2015).

The increased production of dystrophin resulted in increased motility of mice treated with genetically corrected DMD-U7 TIM cells that approached the motility of WT mice; in contrast motility of mice treated with WT TIM cells improved only slightly, as expected with a single intra-muscular injection.

It may be argued that robust expression of dystrophin was observed only with intra-muscular transplantation while the selling point of MABs over satellite cell-derived myogenic progenitors, is their ability to cross the vessel wall and thus be delivered systemically. However, human myogenic cells have difficulties in binding to mouse endothelium and do not engraft efficiently in mouse muscle when delivered systemically (Meng et al, 2011). Indeed, we observed a significant reduction of dystrophin production when cells were delivered intra-arterially rather than intra-muscularly, and this is why in previous work, both pre-clinical and clinical, we performed multiple, consecutive injections (Sampaolesi et al, 2003, 2006; Tedesco et al, 2011, 2012; Cossu et al, 2015). Remarkably, we detected dystrophin production, though at much lower level, even with a single intra-arterial injection of genetically corrected DMD-U7 hMABS.

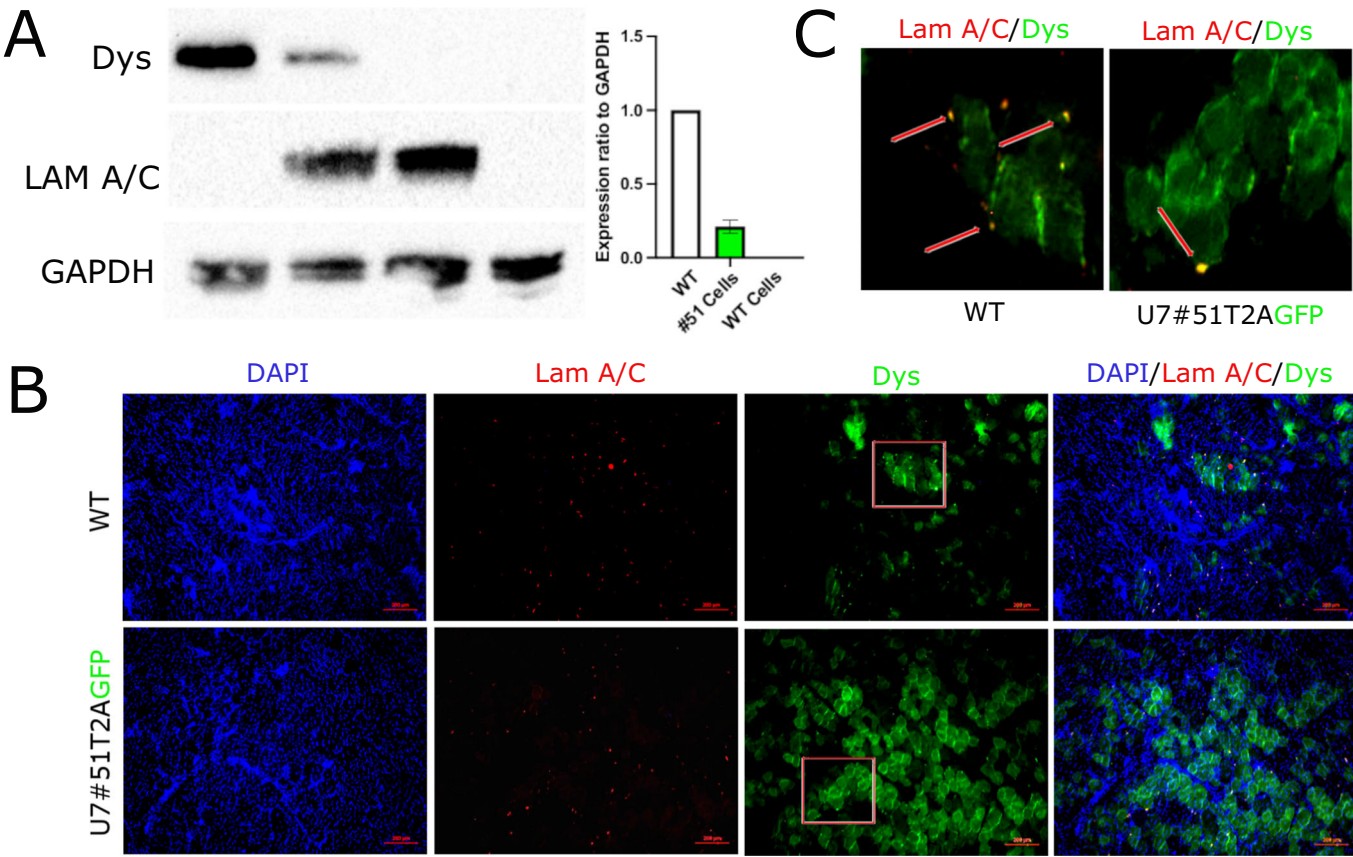

**Figure 5. In vivo Dystrophin protein expression induced by lentiviral vector U7#51T2AGFP after intra-arterial injection.**

(A) Representative WB image of Dys expression from the TA of NSG-mdx-Δ51 or NSG WT mice. Samples are from left to right: 1: NSG WT muscle; 2: NSG-mdx-Δ51 muscle injected with $5\times10^5$ DMD-U7 hMABs; 3: NSG-mdx-Δ51 muscle injected with $5\times10^5$ WT hMABs; 4: NSG-mdx-Δ51 muscle non injected. On the right quantification of Dys expression ($n = 5$ biological replicates). (B) Representative IF images showing expression of Dystrophin and Lamin AC of transverse sections of TA of NSG-mdx-Δ51 mice injected either with $5\times10^5$ WT hMABs. Or with $5\times10^5$ DMD-U7 hMABs ($n = 5$ biological replicates). (C) Expanded view of representative IF images showing expression of Dystrophin and Lamin AC of transverse sections of TA of NSG-mdx-Δ51 mice ($n = 5$ biological replicates). Data Information: Data are shown as mean ± SD. $t$-test are followed by Bonferroni correction unpaired $t$-test. Experiments have been replicated for at least three times. Source data are available online for this figure.

Thus, even if a significantly lower production of dystrophin will be obtained in all muscles downstream of the injected artery, cell administration may and will be repeated several times (Cossu et al, 2015), something nearly impossible with AAV gene therapy. Indeed, a phase I/II trial based upon intra-muscular injection of autologous hMABs, genetically corrected to express U7 snRNA, is running at the time of writing. Furthermore, the heart will remain to be treated, and for this we are developing an alternative strategy based upon administration of mesoangioblasts converted to cardiomyocytes by transient expression of cardiac transcription factors (Efe et al, 2011).

In conclusion we believe this work shows that is possible to produce "therapeutic levels" of dystrophin. Combined with an improved protocol that may enhance other aspect of the transplantation procedure, this strategy may approach or even reach clinical efficacy. Depending upon the outcome of the currently running trial, we would then plan a phase II trial in very young patients, based upon intra-arterial systemic administration of autologous, genetically corrected cells. While it may be argued that what we report here is just an incremental step, is

should be remembered that an increment above a critical threshold makes the difference between success or failure of a novel therapy.

## Methods

### Cells

TIM cells were maintained in 80% DMEM/20% RPMI199 (Sigma), 20% FBS, 25 mg/mL Dexam (Gibco), 10 mg/mL gentamicin (Sigma), 25 mg/mL fetuin (Gibco) and 1x insulin-transferrin-selenium-X (Gibco), supplemented with 5 μg/mL hFGF and 500 μg/mL recombinant hEGF.

hMabs were maintained in Megacell DMEM, 5% FBS, 0.2% β-mercaptoethanol (Gibco), 1% L-Glut (Gibco), 1% Pen/Strep (Gibco) and 1% Non-essential amino acids (Sigma).

MyoD-ER(T) Fibroblast were maintained in DMEM (Sigma), 20% FBS, 1% L-Glut (Gibco), 1% Pen/Strep (Gibco), 1 mM sodium pyruvate (Gibco).

TIM cells were differentiated on collagen (Gibco)-coated culture dishes using DMEM (Sigma), 10 mg/mL gentamicin (Sigma),1x insulin-transferrin-selenium-X (Gibco) while hMabs and MyoD-ER(T) Fibroblast were differentiated on collagen-coated culture dishes using DMEM (Sigma), 4% HS, 1% L-Glut (Gibco) and 1% Pen/Strep (Gibco). Both media were supplemented with 1.25 mM Forskolin (Sigma) to enhance myotube fusion.

## Lentivector

The lentiviral vector used, named U7#51T2AGFP, was derived from pCDH.EFK.MCH.T2A.GFP, (described as T2AGFP) allowing co-expression of the reporter gene GFP and snRNA under the same promoter Human Elongation Factor-1 alpha (EF-1α) (Fig. EV1).

## RT-PCR and qRT-PCR

Total RNA was extracted using TRIzol (Invitrogen) according to the manufacturer's instruction. RNA was reverse transcribed using cDNA preparation kit (Thermoscientific).

For RT-PCR reaction the sequences of the primer pairs were as follow. hGAPDH: Fw 5′-CCACCACCCTGTTGCT-3′, Rev 5′-ACC ACAGTCCATGCCATC-3′; hMyHC: Fw 5′- GGCCAAAATCAA AGAGGTGA-3′, Rev 5′-CGTGCTTCTCCTTCTCAACC-3′; hDYS: Fw 5′-AGCAGTTCAAGCTAAACAACC-3′, Rev 5′-CAAGAGGC ATTGATATTCTCTTG-3′.

For qRT-PCR reaction the sequences of the primer pairs were as follow. hGAPDH: Fw 5′-CCACCACCCTGTTGCT-3′, Rev 5′-ACC ACAGTCCATGCCATC-3′; hMyHC: Fw 5′- GGCCAAAATCAA AGAGGTGA-3′, Rev 5′-CGTGCTTCTCCTTCTCAACC-3′; GFP: Fw 5′-TGATCGGCGACTTCAAGGTG-3′, 5′-CGTTGCTGCGG ATGATCTTG-3′. hDYS 47/52: Fw 5′-TCTCCAGTGGATAAAGG-CAACAA -3′, hDYS 53 Rev 5′-GCCTCCGGTTCTGAAGGTG-3′.

The cDNA for sequencing was extracted according to the instruction provided with QIAquick gel extraction kit (Qiagen), and sequence was performed by GeneWiz.

## FACS analysis

FACs analysis on the EGFP positive cells was performed using a BD LSR Fortessa Special order.

## Immunofluorescence

For immunofluorescence staining of cells, primary antibodies used were: anti-MyHC MF-20 (Development Studies Hybridoma Bank); anti-Dys MANDRA17 (Development Studies Hybridoma Bank); anti-GFP (Ab-13970 Abcam); Secondary antibodies used were: anti-mouse Alexa Fluor 546 and anti-chicken Alexa Fluor 488 (Abcam). Nuclei were stained with DAPI (D9542 Sigma).

For immunofluorescence staining of injected and not injected TA muscles. Tissues were dissected, frozen in liquid nitrogen-cooled isopentane and cut on cryostat (LEICA CM 1850, Leica, Germany) into transverse and longitudinal cryostat sections. Primary antibodies used, in addition to those mentioned above, were: anti-dystrophin DMD (Sigma), anti α-sarcoglycan (HPA 007537 Atlas antibodies), anti-nNOS (PA3-032A R&D System) and anti-LAM A/C (MA3-1000 Invitrogen). Secondary antibodies used

were: anti-mouse Alexa Fluor 488, anti-rabbit Alexa Fluor 546 (Abcam). Nuclei were stained with DAPI (D9542 Sigma).

## Histology

Haematoxylin and Eosin staining was performed according to the instruction provided by Sigma Aldrich (Haematoxylin H9627 and Eosin 230251, Sigma). Sirius Red staining was performed according to the instruction provided Picro Sirius red stain kit (Ab 150681, Abcam).

## In situ

The In situ assay was performed according to the instruction provided with ACDbio RNAscope Multiplex Fluorescent Reagent Kit. The signal was amplified using specific probe for GFP, LAM A/C and the U7 snRNA detection, specifical design for us by ACDbio.

## Western blot

Cells were lysed in RIPA buffer (10 mM TRIS, 100 mM NaCl, 1 mM EDTA, 1% Triton, 10% Glycerol, 0.1% SDS and 1% protease inhibitor). Protein concentration was determined with the Bio-Rad Protein Assay. Proteins were separated in 4–15% gradient, pre-casted SDS PAGE and then transferred onto a nitrocellulose membrane using standard protocol. The blots were incubated with the following antibodies: anti-MyHC MF-20 (Development Studies Hybridoma Bank); anti-Dys MANDRA17 (Development Studies Hybridoma Bank); anti-GAPDH (Ab 125247 Abcam) and anti-LAM A/C (MA3-1000 Invitrogen). Proteins were visualized by an enhance chemiluminescence method (Thermoscientific) according to the manufacturer's instructions.

## In vivo intra-muscular injection

The use of animals in this study was authorized (license PDB0CF0C2); mice were fed ad libitum and allowed continuous access to tap water. NGS mice carrying a skippable mutation of exon 51 (NOD.Cg-*Prkdc*$^{scid}$ *Il2rg*$^{tm1Wjl}$ Tg(HLA-A/H2-D/B2M)1Dvs/SzJ) obtained from Jackson Laboratory (USA). Mice (6 per experimental group) were intramuscularly injected into *Tibialis Anterior* muscle (TA) and/or *Gastrocnemius* (G) and/or *Quadriceps* (Q) with either $5 \times 10^5$ TIM cells or hMABs in 50 μL of PBS, unless otherwise specified; contra-lateral muscles were used as not injected control ones. Mice were culled after 30 days since injection, unless otherwise specified.

## In Vivo subcutaneous injection

Mdx-SCID mice were subcutaneously injected in the back with $5 \times 10^5$ TIM cells mixed in a Matrigel (Corning) plug (diluted 1:1 with the cell suspension).

## In vivo intra-arterial injection

5 NSG-mdx-Δ51mice were intra-arterially injected with $5 \times 10^5$ DMD-U7 hMABs or wt hMABs in 50 μL of PBS + Heparin as described (Sampaolesi et al, 2003). One month after the injection

mice were culled. *Tibialis Anterior* muscles (TA) were collected for IF and *Gastrocnemius* (G) were collected for WB assay.

## Motility assay

Twelve males NOD.Cg-*Prkdc*$^{scid}$ *Il2rg*$^{tm1Wjl}$ Tg(HLA-A/H2-D/B2M) 1Dvs/SzJ of 6 months of age split into four groups (3 WT mice, 3 DMD mice transplanted with WT TIM cells, 3 DMD mice transplanted with DMD-U7 cells, 3 DMD untreated mice). Mice were intramuscularly injected into the TA, Gas, Quad, both the lower limbs with $0.5 \times 10^6$. Thirty days after the transplantation the animals were challenged on a treadmill recording the distance and running time with the following protocol: 2 min warm up at start 8 m/min; 1 min gradually increasing to 10 m/min followed by 10 m/min to a max 15 m/min over 10 min. Their activity was monitored continuously.

## Statistical analysis

All the in vitro experiments were repeated no less than three times, each in triplicate, to ensure to measure statistically significant differences upon different experimental treatments. For cross-correction experiments, we quantified the amount of protein expressed by densitometric analysis of WB and compared the different samples by *t*-test testing with GraphPad Prism 9.0 software.

In vivo experiments were conducted on experimental groups of 6 animals. This represents the best compromise between the 3R guidelines and the need to have reproducibility and consistency in the results. All the in vivo data were analysed with GraphPad Prism 9.0 software. For comparison and analysis between the different animal *t*-test analysis was used.

The sample size was selected based on the understanding of a biologically meaningful effect and the variability in the primary endpoints. Previous experimental data were used to assess both aspects. Our experience indicates that groups of five animals are the minimal to predict possible significance of results, provided that the difference between treated and controls is large and variability modest. Data analysis was conducted according to a pre-specified statistical analysis plan. Based on previous experimental data, we used Z-score 1.96 for 95% confidence, population standard deviation 0.5, effect size 0.5. and we calculated we need a minimum sample size of approximately 4 to achieve a 95% confidence level for each group using both ANOVA and *t*-test.

## Data availability

This study includes no data deposited in external repositories.

## Peer review information

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

## Acknowledgements

This work was supported by MRC grants MR/P016006/1 and MR/S015116/1, Duchenne Parent Project Italy, ERC-2019-ADG 884952 and Wellcome Trust HICF 107572 to GC and ERC-2019-SyG 855923-ASTRA, AIRC IG 2019 Id. 23053 and "National Center for Gene Therapy and Drugbased on RNA Technology" (CN00000041), NextGenerationEU PNRR MUR to IB. The production of the NOD.Cg-Prkdc Il2rg Tg(HLA-A/H2-D/B2M)1Dvs/SzJ mice was initially supported by MRC CiC grant to the University of Manchester.

## Author contributions

**Francesco Galli**: Investigation. **Laricia Bragg**: Investigation. **Maira Rossi**: Investigation. **Daisy Proietti**: Investigation. **Laura Perani**: Investigation. **Marco Bacigaluppi**: Investigation. **Rossana Tonlorenzi**: Investigation. **Tendai Sibanda**: Investigation. **Miriam Caffarini**: Investigation. **Avraneel Talapatra**: Investigation. **Sabrina Santoleri**: Investigation. **Mirella Meregalli**: Investigation. **Beatriz Bano-Otalora**: Investigation. **Anne Bigot**: Investigation. **Irene Bozzoni**: Resources. **Chiara Bonini**: Resources. **Vincent Mouly**: Resources. **Yvan Torrente**: Resources. **Giulio Cossu**: Conceptualization.

## Disclosure and competing interests statement

The authors declare that they have no competing interests. Prof Giulio Cossu is an Editorial Advisory Board Member. This has no bearing on the editorial consideration of this article for publication.

# Expanded View Figures

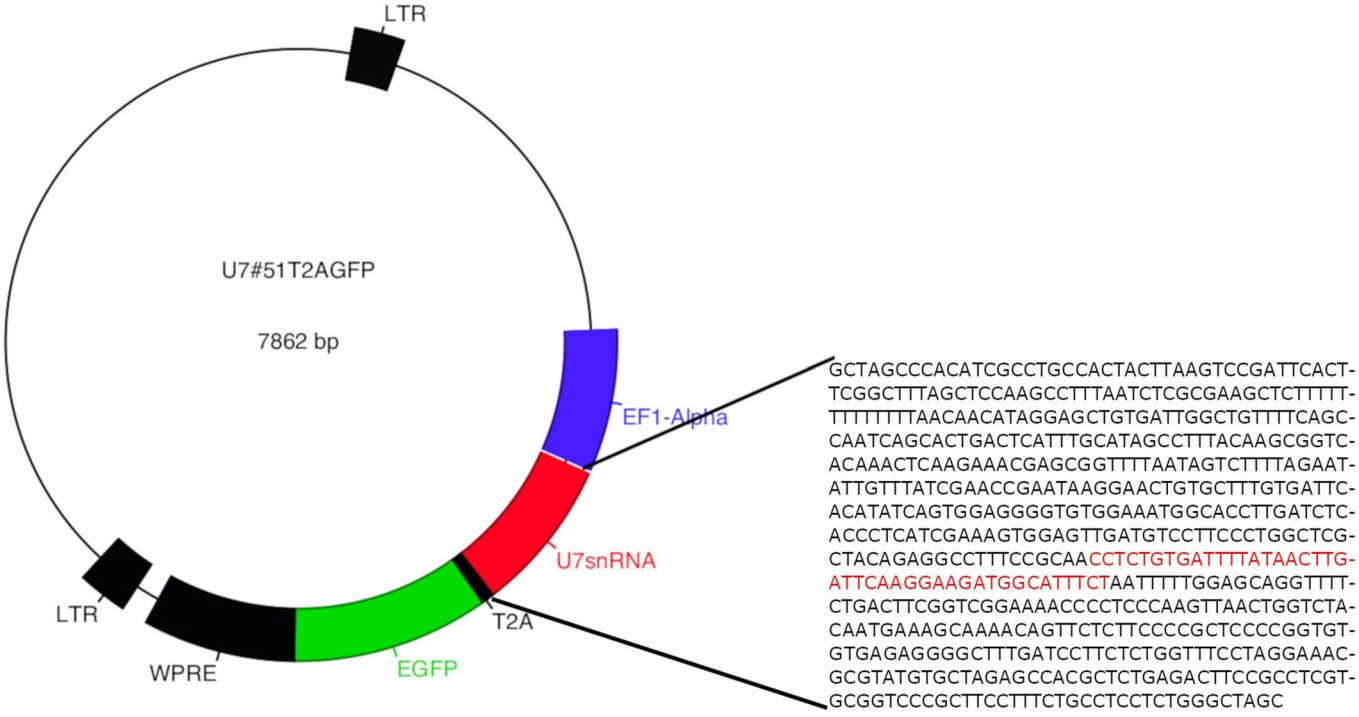

GCTAGCCCACATCGCCTGCCACTACTTAAGTCCGATTCACT-
TCGGCTTTAGCTCCAAGCCTTTAATCTCGCGAAGCTCTTTTT-
TTTTTTTTAACAACATAGGAGCTGTGATTGGCTGTTTTCAGC-
CAATCAGCACTGACTCATTTGCATAGCCTTTACAAGCGGTC-
ACAAACTCAAGAAACGAGCGGTTTTAATAGTCTTTTAGAAT-
ATTGTTTATCGAACCGAATAAGGAACTGTGCTTTGTGATTC-
ACATATCAGTGGAGGGGTGTGGAAATGGCACCTTGATCTC-
ACCCTCATCGAAAGTGGAGTTGATGTCCTTCCCTGGCTCG-
CTACAGAGGCCTTTCCGCAACCTCTGTGATTTTATAACTTG-
ATTCAAGGAAGATGGCATTTCTAATTTTTGGAGCAGGTTTT-
CTGACTTCGGTCGGAAAACCCCTCCCAAGTTAACTGGTCTA-
CAATGAAAGCAAAACAGTTCTCTTCCCCGCTCCCCGGTGT-
GTGAGAGGGGCTTTGATCCTTCTCTGGTTTCCTAGGAAAC-
GCGTATGTGCTAGAGCCACGCTCTGAGACTTCCGCCTCGT-
GCGGTCCCGCTTCCTTTCTGCCTCCTCTGGGCTAGC

**Figure EV1. Map of I lentiviral vector U7#51T2AGFP and sequence of the snRNA.**

Schematic overview of the lentivector U7#51T2AGFP derived from pCDH.EFK.MCH.T2A.GFP, and sequence of the U7 snRNA with underlined in red the antisense sequence. Source data are available online for this figure

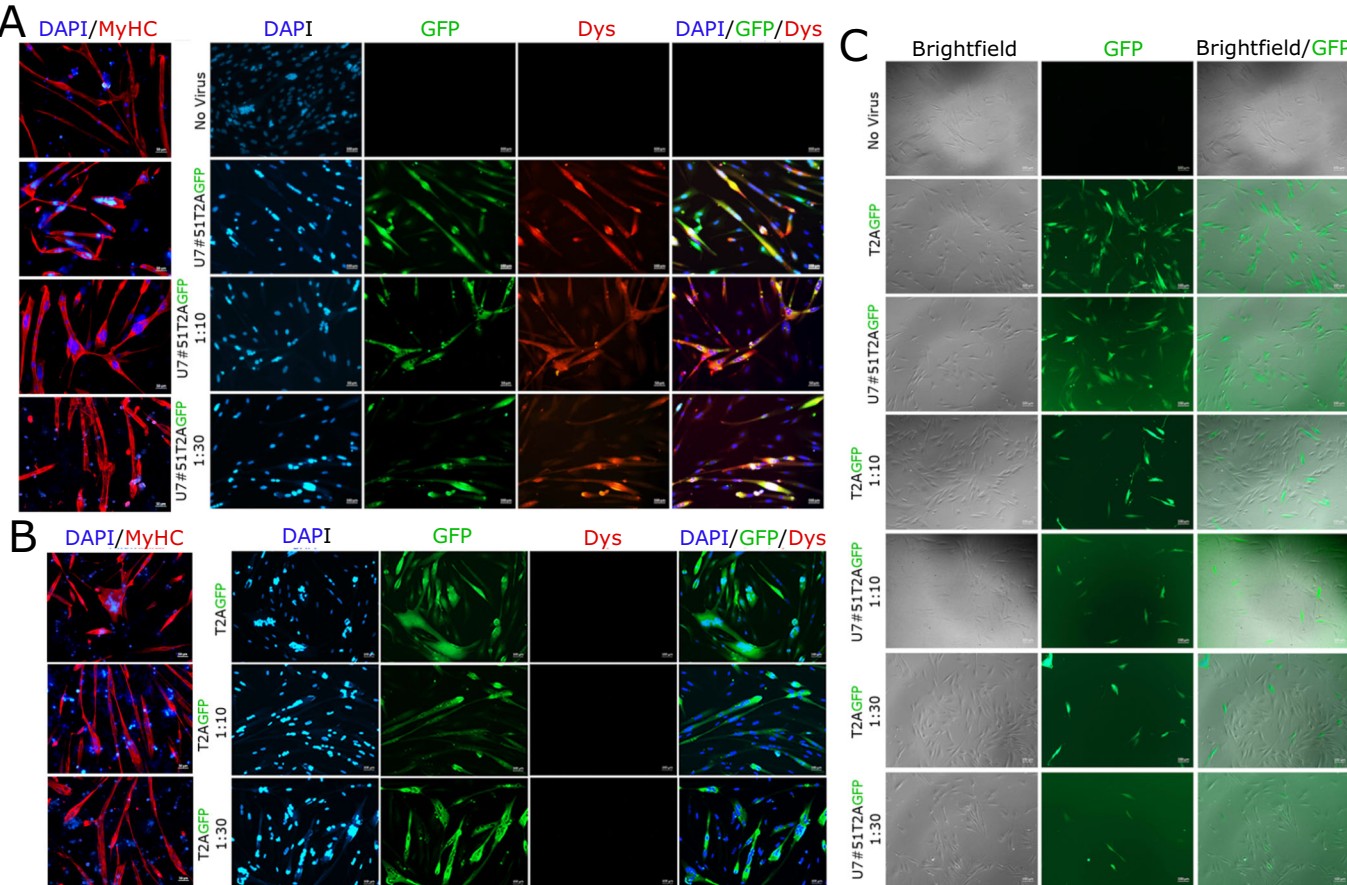

**Figure EV2.  Cell-mediated exon skipping induced by lentiviral vector U7#51T2AGFP.**

(**A**) Representative IF images showing expression of Dystrophin and GFP in a DMD-U7/DMD TIM cells co-culture ($n = 3$ biological replicates). (**B**) Representative IF images showing expression of Dystrophin and GFP in a DMD-GFP/DMD TIM cells co-culture ($n = 3$ biological replicates). (**C**) Representative live analysis showing expression of GFP in a DMD-U7 or DMD-GFP/DMD TIM cells co-culture ($n = 3$ biological replicates). Data Information: Experiments have been replicated for at least three times. Source data are available online for this figure

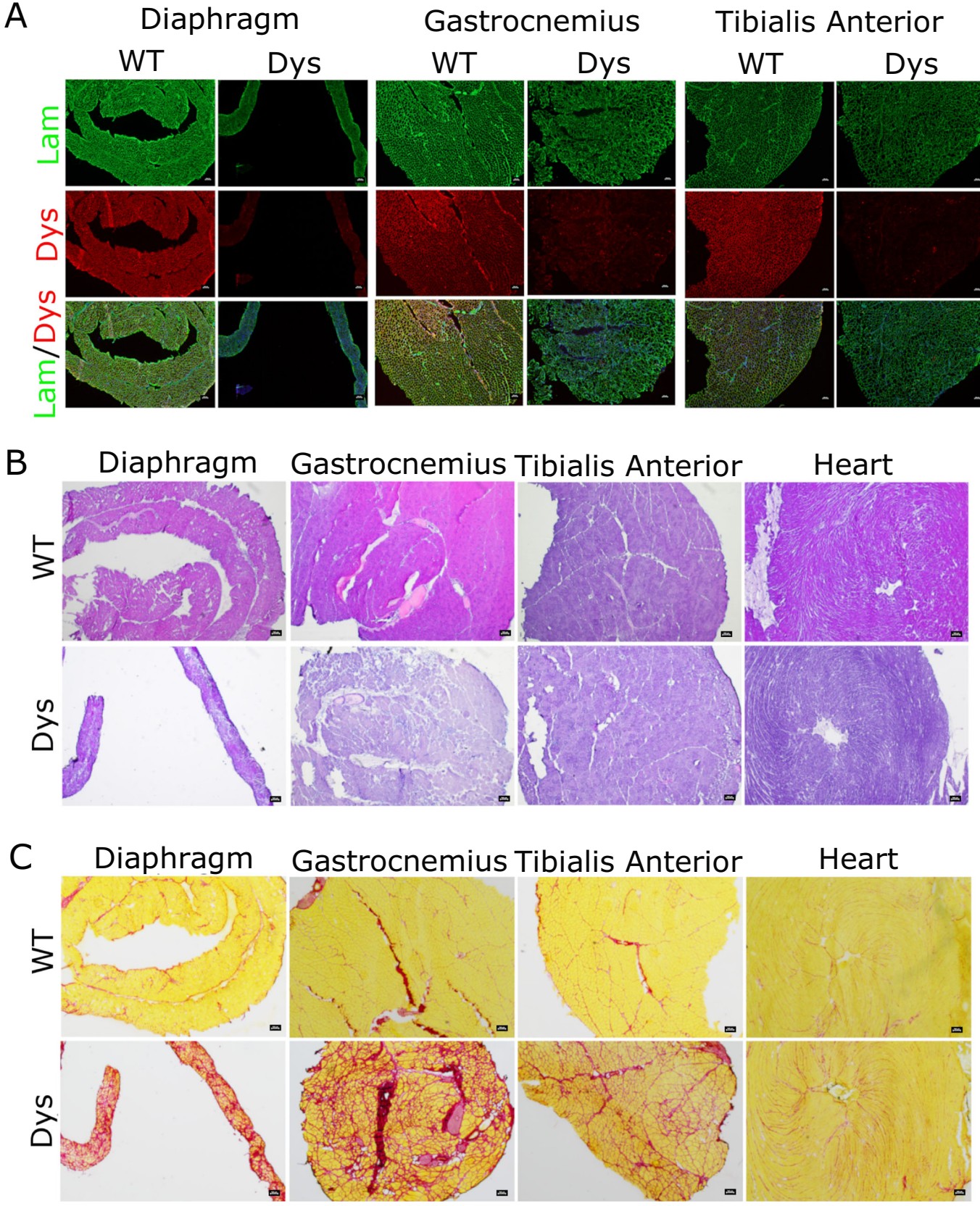

◄ **Figure EV3. Phenotypical characterization of NSG-mdx-D51 mice.**

Phenotypical characterization of 6 months old NSG-mdx-D51 mice by (A). Representative IF images showing expression of dystrophin and Laminin of Diaphragm; Gastrocnemius; Tibialis Anterior ($n = 6$ biological replicates). (B) Representative Haematoxylin and Eosin staining of Diaphragm; Gastrocnemius; Tibialis Anterior; Heart. ($n = 6$ biological replicates). (C) Representative Sirus Red staining of Diaphragm; Gastrocnemius; Tibialis Anterior; Heart ($n = 6$ biological replicates). Data Information: Experiments have been replicated for at least three times. Source data are available online for this figure

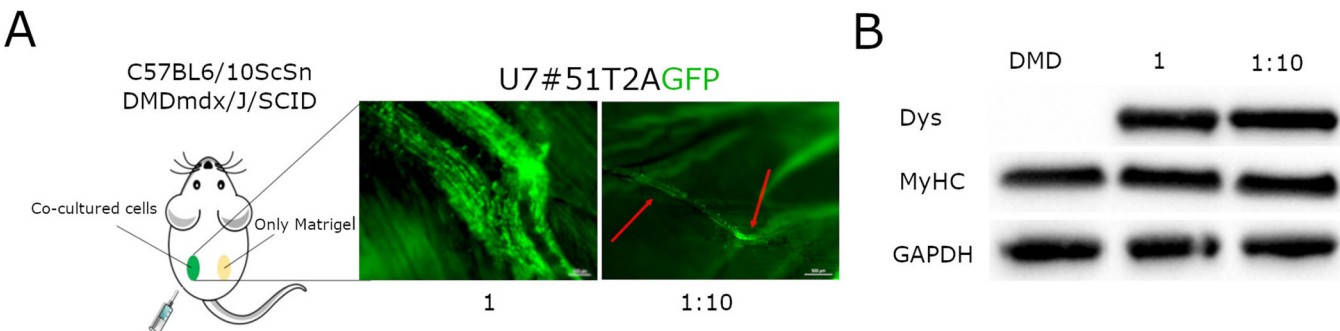

**Figure EV4.  Transplantation of DMD-U7 cells in a Matrigel plug under the back skin of mdx/SCID mouse.**

(A) Representative IF images showing expression of GFP in a DMD-U7/DMD TIM cells co-culture in a matrigel plug implanted under the dorsal skin of mdx/SCID mice (*n* = 3 biological replicates). (B) Representative WB image of Dys expression from matrigel plug implanted under the dorsal skin of mdx/SCID. Samples are from left to right: DMD: DMD TIM cells; 1: DMD U7 cells; 1:10: DMD U7 cells co-cultured with a 10-fold excess of DMD TIM cells. (*n* = 3 biological replicates). Data Information: Experiments have been replicated for at least three times. Source data are available online for this figure

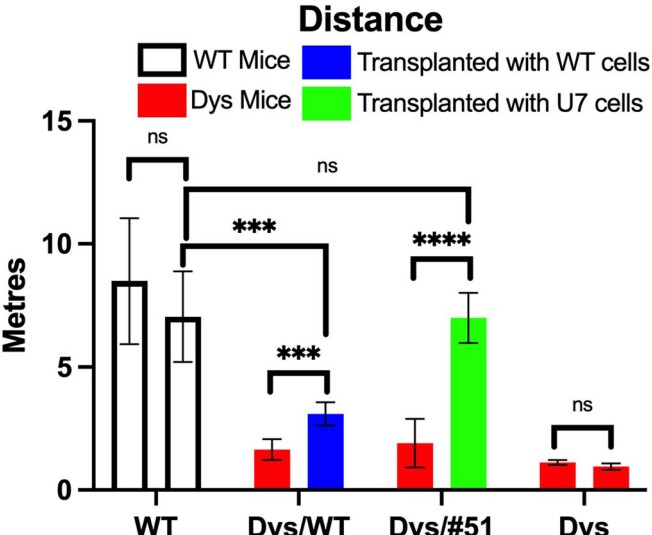

**Figure EV5.   Improving of mice motility after cells transplant.**

Treadmill motility assay analysis before and after transplantation with either $5 \times 10^5$ WT TIM or $5 \times 10^5$ DMD-U7 cells ($n = 6$ biological replicates). Data Information: Data are shown as mean ± SD. *$P < 0.05$, **$P < 0.01$, ***$P < 0.001$, ****$P < 0.0001$. Multiple $t$-test are followed by Bonferroni correction unpaired $t$-test. The mean differences between two group were calculated with 2-way anova. Experiments have been replicated for at least three times. Source data are available online for this figure

