## [Peer Review File · EMBO Molecular Medicine]

Cell-mediated exon skipping normalizes dystrophin expression and muscle function in Duchenne Muscular Dystrophy.

Francesco Galli, Laricia Bragg, Maira Rossi, Daisy Proietti, Laura Perani, Marco Bacigaluppi, Rossana Tonlorenzi, Tendai Sibanda, Miriam Caffarini, Avraneel Talapatra, Sabrina Santoleri, Mirella Meregalli, Beatriz Bano-Otalora, Anne Bigot, Irene Bozzoni, Chiara Bonini, Vincent Mouly, Yvan Torrente, and Giulio Cossu

Corresponding authors: Francesco Galli (francesco.galli@manchester.ac.uk) , Giulio Cossu (giulio.cossu@manchester.ac.uk)

Review Timeline:

Submission Date:	27th Feb 23
Editorial Decision:	12th Apr 23
Appeal:	21st Apr 23
Editorial Decision:	10th May 23
Revision Received:	20th Nov 23
Editorial Decision:	21st Dec 23
Revision Received:	12th Jan 24
Accepted:	22nd Jan 24

Editor: Zeljko Durdevic

Transaction Report:

12th Apr 2023

Decision on your manuscript EMM-2023-17623

Dear Dr. Galli,

Thank you for the submission of your manuscript to EMBO Molecular Medicine, and please accept my apologies for the delay in getting back to you. We have now received feedback from two of the three reviewers who agreed to evaluate your manuscript. As the referee #2 will unfortunately not be able to return his/her report in a timely manner, and given that both reviewers provide very similar recommendations, we prefer to make a decision now in order to avoid further delay in the process.

As you will see from their reports pasted below, while they recognize interest of your study, they also raise serious concerns, particularly regarding, but not limited to, the limited conceptual advance and low medical impact. As clear and conclusive insight into a novel, clinically relevant observation is crucial for publication in EMBO Molecular Medicine, and together with the fact that we only accept papers that receive enthusiastic support upon initial review, I am afraid that we cannot offer to consider the manuscript further.

I am sorry that I could not bring better news this time and hope that the referee comments are helpful in your continued work in this area.

Yours sincerely,

Zeljko Durdevic

***** Reviewer's comments *****

Referee #1 (Comments on Novelty/Model System for Author):

- 1- Quantification for many experiments is missing, and in some cases, it appears the graph represents only one sample. The "n" per group is not clearly described (one needs to search in M&M). This information should be provided in Figure Legends.
- 2- Mesoangioblasts are not novel and the lentiviral vector expressing the U7 snRNA was first described 20 years ago, as mentioned in the introduction (De Angelis et al., PNAS, 2002).
- 3- There is potential for impact but considering all the issues described in detail below, for now, medical impact is medium.

Referee #1 (Remarks for Author):

Francesco Galli, Giulio Cossu and colleagues report a cell-mediated exon skipping protocol to treat Duchenne Muscular Dystrophy (DMD). Given the limited engraftment encountered in recent clinical trials assessing transplantation of mesoangioblasts in DMD patients, they postulate that cell therapy wasn't (and won't) be effective in DMD due to the fibrotic state of DMD muscles. Based on this premise, they developed a therapy in which minimal cell engraftment may lead to some therapeutic benefit through cross-correction of resident dystrophic nuclei by the U7 small nuclear RNA engineered to skip exon 5 of the dystrophin gene. While the study is of interest, there are numerous concerns with this manuscript in its present form.

Major comments:

- 1- The introduction is confusing. Cossu and colleagues have published a body of literature on the regenerative potential of mesoangioblasts, which led to a first-in-human clinical trial. Because this trial resulted in limited engraftment (<1%), which could have been for several reasons, including low cell dose and age of patients at enrollment (as suggested by the authors), but also due to inefficient immunosuppression and/or lack of regenerative potential of injected cells, the authors postulate in the introduction that "even starting at an earlier age and implementing the protocol, engraftment will always remain low in tissues such as muscle and brain where ablation of diseased resident cells is not possible". There is no scientific basis for this statement, nor for the comparison between muscle and brain. The reference listed is a review by Thomas Rando's group that does not support this claim. In addition, this statement goes against all the literature published by this group in the last 15 years promising to treat muscular dystrophy with mesoangioblasts. Oddly, in the discussion, the authors make overpromising claims again, such as "this work is ground-breaking as we can now produce "therapeutic levels of dystrophin", and their intent to cure young patients.

2- Key information is missing. The authors mentioned the peak of degeneration in this novel mouse model is at one month of age but there is no information about the timing of transplantation. The levels of fibrosis are also not clear. This is critical given the authors claim that poor engraftment was due to advanced fibrotic pathology. The authors also do not explain whether the mice were subjected to any type of pre-injury prior to transplantation - was cardiotoxin or other injury performed? What are the levels of engraftment? It is necessary to show clear quantification of engraftment (IF and western blot). Are engraftment levels lower in this mouse model, opposed to previous studies involving the transplantation of mesoangioblasts? Are engraftment levels low because cells were injected in fibrotic muscle? Differences and commonalities between previous and current studies should be acknowledged.

3- Overall, figure organization and their respective descriptions in the text is chaotic. For instance, not all figures are described, and in many instances, they are covered out of order (for example, Fig 1C is described before 1A).

4- Figure 2B: It is important to demonstrate whether there are differences in engraftment between wt/DMD and DMDu7/DMD. For some samples, there are no error bars suggesting that they quantified only one gel. Protein expression results are more relevant than the RNA data shown in Figure 1. Better attention to rigor is required.

5- Figure S3A: In addition to merged images, the authors should include images for laminin and Dys staining separately as it is not easy to visualize Dys staining in the merge. The TA staining looks the same between wt and mdx51 mice. No information is provided regarding the age of the mice analyzed in this figure. The authors should provide images with better resolution and higher magnification.

6- Fig 3A and 3B: The authors quantify the western blot but find no significant difference between DMD and wt. What does this mean? It is claimed that DMD u7 cells cross-correct resident dystrophic nuclei to produce greater quantities of dystrophin but the data seems to be saying something different.

7- It is difficult to find the number of mice used in different experiments. This information should be included in each figure legend.

8- Fig 4: The authors should include a negative control for the sn U7 staining so the reader can appreciate what the background level is.

Minor comments:

- Results describing cell plug as described in Fig. S5, not Fig. S4 as described in the text.

Referee #3 (Comments on Novelty/Model System for Author):

1. The in vivo experiments were on just 6 animals per treatment group which is low and could results in type I and type II statistical errors. Power analysis was not reported in the study. There are concerns with experimental rigor including were those scoring experimental outcomes blinded to the treatment groups.

2. Novelty is medium as similar approaches using exon skipping in myogenic cells have been reported previously

3. Medical impact is low since this approach will probably not treat DMD related dilated cardiomyopathy, the primary cause of death in DMD

4. The model system is adequate and the development of a humanized DMD mouse is a strength of the manuscript

Referee #3 (Remarks for Author):

This manuscript explores the use of a cell-mediated exon skipping strategy to restore dystrophin expression in myogenic cells and a humanized mouse model of Duchenne muscular dystrophy (DMD). The authors report that co-culture of corrected dystrophin deficient cells with dystrophin negative cells increased dystrophin production. They report that this occurs through transmission of exon skipping from corrected nuclei to nearby nuclei in muscle. They conclude this form of cell-mediated exon skipping may be usually in the treatment of DMD.

This could be a potentially an interesting manuscript, but there are several major concerns that need to be addressed.

1. In vivo experiments of 6 animals per treatment group is low and could results in type I and type II statistical errors. Power analysis should be used to determine experimental animal numbers for the study, and this reported within the manuscript.

2. Were those scoring experimental outcomes blinded to the treatment groups and how was genetic drift managed in the mouse colonies?

3. Fig 1D: Why is dystrophin (red) localized within the cytoplasm of the muscle cells and not at the sarcolemma?

4. Fig 2A: There is considerable red background staining, so it is difficult to conclude if intracellular diffusion of the snRNA from a

GFP positive nucleus to an adjacent nucleus occurs. Several closer nuclei to the GFP positive nucleus are completely negative so it may be helpful to quantify GFP/snRNA positive vs non-GFP/snRNA positive nuclei.

5. Fig 2B-D: GAPDH levels change in dystrophic muscle compared to unaffected muscle. In Fig 2 GAPDH levels are highly variable between the lanes.
6. There is a concern that Fig 2D that GAPDH band is overexposed. The current accepted approach is to normalize dystrophin to total proteins in the same lane rather than a single housekeeping protein.
7. Fig 2B-D: Western blots using anti-dystrophin antibodies against the N-terminus, mid-region and C-terminus of the dystrophin protein would be helpful in assessing full length vs truncated dystrophin.
8. Fig 3: The dystrophin expression is not uniform, but patchy within the muscle. The authors should comment on how regional differences in dystrophin expression using this technique could affect therapeutic potential.
9. Fig 4D: The western panel for dystrophin is highly overexposed compared to the control panels. This suggests very low levels of dystrophin are achieved and the conclusion that levels are the same at 1 month after injection based on these blots are a concern.
10. Motility assays are in Fig 4C (not Fig 3C as in the manuscript text)
11. The authors should use the ex vivo muscle contractile to assess muscle force generation.
12. DMD patients die prematurely from respiratory failure and dilated cardiomyopathy. The authors should comment on how this approach can be used to treated cardiopulmonary dysfunction.

As a service to authors, EMBO provides authors with the possibility to transfer a manuscript that one journal cannot offer to publish to another EMBO publication. The full manuscript and if applicable, reviewers reports are automatically sent to the receiving journal to allow for fast handling and a prompt decision on your manuscript. For more details of this service, and to transfer your manuscript to another EMBO title please click on Link Not Available

Dr. Zeljko Durdevic
Scientific Editor
EMBO Molecular Medicine

April 21, 2023.

Dear Dr Durdevic,

Thank you for the time and the effort needed to review our ms. Though disappointed, we perfectly understand that, as an Editor, with 2/2 negative or at most lukewarm comments you had no other choice than reject our ms.

We took a few days to carefully evaluate the Referees comments. We admit and regret that the ms contains a number of imprecisions, but these could be corrected in one day. The real issue is the statement by the Referees that the paper is not novel, has medium impact and is far from clinical translation. Neither Referee discusses the only real advance of the work: i.e. that fact that the snRNA enters neighbouring nuclei of the regenerating muscle fibre with which the transplanted cell fuses. In this way exon skipping is amplified and leads to dystrophin production that ranges between 30-50% of the level of a healthy muscle, well in the therapeutic range (Anthony et al. 2014 doi: 10.1212/WNL.0000000000001025). This result was repeated on three different myogenic cell types in vitro and in vivo. Moreover, we observed, for the first time in 25 years, an almost "back to normal" functional recovery on mice transplanted with DMD, corrected cells. Together these data strongly suggest that, with all the caveat of scaling up to patients and to systemic distribution, this approach may lead to an efficacious therapy.

In view of the lack of attention to the actual data presented in contrast with criticisms on the presentation of the data (see itemised replies), we would request that the ms is sent to one or two more Referees, one of whom ideally may be in the list of leading scientists in the field that we submitted.

With best wishes,

Francesco Galli

Francesco Galli

Giulio Cossu

Referee #1 (Comments on Novelty/Model System for Author):

1- Quantification for many experiments is missing, and in some cases, it appears the graph represents only one sample. The "n" per group is not clearly described (one needs to search in M&M). This information should be provided in Figure Legends.

All the WB are quantified so that this first criticism lacks substance. Only in one experiment Fig 2C, we have only one sample because we had access to a muscle biopsy of only one DMD patient with the specific mutation needed. We apologise for not having indicated "n" in the Figure legend and will do this (though not required by editorial policy), but it is very different from non-indicating the replicates at all.

2- Mesoangioblasts are not novel and the the lentiviral vector expressing the U7 snRNA was first described 20 years ago, as mentioned in the introduction (De Angelis et al., PNAS, 2002).

In this comment the Referee misses the point that the use of a cell mediated exon-skipping brings dystrophin production to an unprecedented therapeutic level (other than with AAV) and this is novel and potentially ground-breaking.

3- There is potential for impact but considering all the issues described in detail below, for now, medical impact is medium.

This is the Referee opinion, probably because she/he disregarded the novel mechanism and the level of dystrophin expression, not the least relevant point for DMD therapy.

Referee #1 (Remarks for Author):

Francesco Galli, Giulio Cossu and colleagues report a cell-mediated exon skipping protocol to treat Duchenne Muscular Dystrophy (DMD). Given the limited engraftment encountered in recent clinical trials assessing transplantation of mesoangioblasts in DMD patients, they postulate that cell therapy wasn't (and won't) be effective in DMD due to the fibrotic state of DMD muscles. Based on this premise, they developed a therapy in which minimal cell engraftment may lead to some therapeutic benefit through cross-correction of resident dystrophic nuclei by the U7 small nuclear RNA engineered to skip exon 5 of the dystrophin gene. While the

study is of interest, there are numerous concerns with this manuscript in its present form.

Major comments:

1- The introduction is confusing. Cossu and colleagues have published a body of literature on the regenerative potential of mesoangioblasts, which led to a first-in-human clinical trial. Because this trial resulted in limited engraftment (<1%), which could have been for several reasons, including low cell dose and age of patients at enrollment (as suggested by the authors), but also due to inefficient immunosuppression and/or lack of regenerative potential of injected cells, the authors postulate in the introduction that "even starting at an earlier age and implementing the protocol, engraftment will always remain low in tissues such as muscle and brain where ablation of diseased resident cells is not possible". There is no scientific basis for this statement, nor for the comparison between muscle and brain. The reference listed is a review by Thomas Rando's group that does not support this claim. In addition, this statement goes against all the literature published by this group in the last 15 years promising to treat muscular dystrophy with mesoangioblasts. Oddly, in the discussion, the authors make overpromising claims again, such as "this work is ground-breaking as we can now produce "therapeutic levels of dystrophin", and their intent to cure young patients.

The review by Thomas Rando (doi: 10.1172/JCI142031) refers to the sentence "Degeneration is followed by regeneration carried out by satellite cells, resident myogenic stem/progenitor cells and, to a minor extent, interstitial cells such as pericytes" and not to what the reviewer states. The reference listed for the "even starting at an earlier age and implementing the protocol, engraftment will always remain low in tissues such as muscle and brain where ablation of diseased resident cells is not possible" is (Cossu G. et al. doi: 10.1016/S0140-6736(17)31366-1). This is a commonly accepted concept in the field and dates to the first years of bone marrow transplantation when myeloablation increased engraftment leading to successful outcomes. It is taught in medical schools.

The last sentence is wrong and misleading. At variance with Companies that continue, trial after trial, to claim efficacy and safety with nothing changing in the progress of the disease, we never promised any therapy; we had encouraging results in animals (Sampaolesi et al. 2003, doi:10.1126/science.1082254; Sampaolesi et al. 2006, doi:10.1038/nature05282), but with multiple, intra-specific or even syngeneic transplantations; based on these results we run a phase I/IIa clinical trial and reported lack of efficacy (Cossu et al. 2015 doi: 10.15252/emmm.201505636.), even though the amount of dystrophin we detected was more than what currently mentioned as "efficacious" in a recent report (Servais et al. 2022, doi: 10.1089/nat.2021.0043.) with no western blot shown (also because the amount reported, 1.09% of a healthy muscle, would not be detectable on WB). Indeed, Referee fails to evaluate the amount of dystrophin and the functional recovery, that are the only novel results that justify a cautious optimism.

2- Key information is missing. The authors mentioned the peak of degeneration in this novel mouse model is at one month of age but there is no information about the timing of transplantation. The levels of fibrosis are also not clear. This is critical given the authors claim that poor engraftment was due to advanced fibrotic pathology. The authors also do not explain whether the mice were subjected to any type of pre-injury prior to transplantation - was cardiotoxin or other injury performed? What are the levels of engraftment? It is necessary to show clear quantification of engraftment (IF and western blot). Are engraftment levels lower in

this mouse model, opposed to previous studies involving the transplantation of mesoangioblasts? Are engraftment levels low because cells were injected in fibrotic muscle? Differences and commonalities between previous and current studies should be acknowledged.

We apologise for the missing information on the timing of the transplantation (about 2 months of age); it will be added in a revised version. The phenotype of this mouse is similar to the classic mdx mouse and the level of fibrosis (Suppl. Fig 3C) is comparable to what reported in mdx.

We do not explain experiments that we did not do: we did not induce injury, e.g with CTX as it confuses original pathology with acute damage and would never be allowed in a patient. The quantification of engraftment was carried out by IF (shown in Fig.3C) using Lam A/C, a specific human marker and it is clearly showed in the graph. Differences with previous papers were due to different protocols (multiple injections, not performed here, use of wt mouse cells in mice, and lack of quantification of engraftment in papers published 20 years ago). We may discuss this in the revised version.

3- Overall, figure organization and their respective descriptions in the text is chaotic. For instance, not all figures are described, and in many instances, they are covered out of order (for example, Fig 1C is described before 1A).

Fig 1C is described before Fig 1A to indicate the level of transduction of the cells, important to explain the results in FIG 1A and the correct percentage of the ratio of transduced cells for the results in Fig 1D. We may rearrange the figure.

4- Figure 2B: It is important to demonstrate whether there are differences in engraftment between wt/DMD and DMDu7/DMD. For some samples, there are no error bars suggesting that they quantified only one gel. Protein expression results are more relevant than the RNA data shown in Figure 1. Better attention to rigor is required.

All the experiment were made in triplicate, and we will add the error bar to Fig 2B. We agree about the relevance of protein expression, but RNA data, shown in figure 1A, show the ability of snRNA to induce the exon skipping and to generate the correct shorter RNA sequence, not to quantify the amount of dystrophin mRNA produced.

5- Figure S3A: In addition to merged images, the authors should include images for laminin and Dys staining separately as it is not easy to visualize Dys staining in the merge. The TA staining looks the same between wt and mdx51 mice. No information is provided regarding the age of the mice analyzed in this figure. The authors should provide images with better resolution and higher magnification.

We can provide separate images for a better visualization and we will add information about the age of the animal used. We will also provide image with better resolution. The choice of the magnification is to show a large area of the muscle and not only a selected area, and so have a broader view of the phenotype; insets at higher magnification may be added.

6- Fig 3A and 3B: The authors quantify the western blot but find no significant difference between DMD and wt. What does this mean? It is claimed that DMD u7 cells cross-correct resident dystrophic nuclei to produce greater quantities of dystrophin, but the data seems to be saying something different.

It means that the U7-mediated gene correction leads to unprecedented accumulation of dystrophin and this is confirmed by the much lower amount of dystrophin produced by muscles transplanted with the same number of wt cells. It approaches 50% of the level of the wt mouse.

7- It is difficult to find the number of mice used in different experiments. This information should be included in each figure legend.

We will provide the information also in the figure legend.

8- Fig 4: The authors should include a negative control for the sn U7 staining so the reader can appreciate what the background level is.

A negative control for the snU7 is shown in Fig S2, where part of the cells was transduced with a control LV expressing only GFP. No dystrophin was detected. We disagree on the relevance of this control for the in vivo experiments. A negative control for the snU7 won't give any information about background level for the simple fact that antibodies do not recognize the snRNA but the product of the exon-skipping, dystrophin protein that we show by IF and WB. If deemed useful, we may add a figure of an untreated dystrophic muscle, stained for nNOS, Dys and LAM A/C

Minor comments:

- Results describing cell plug as described in Fig. S5, not Fig. S4 as described in the text.

Thanks for bringing this to our attention. We will correct in the text.

Referee #3 (Comments on Novelty/Model System for Author):

1. The in vivo experiments were on just 6 animals per treatment group which is low and could result in type I and type II statistical errors. Power analysis was not reported in the study. There are concerns with experimental rigor including were those scoring experimental outcomes blinded to the treatment groups.

As indicated in the paragraph on statistical analysis, the number of 6 animals in in vivo experiments represents the best compromise between the 3R guidelines and the need to have reproducibility and consistency in the results.

We checked recently published papers on the topic and report the first three papers we found:

Sun et al. 2022 DOI: 10.1016/j.stem.2022.03.004: 6 mice/exp group (data in Mat @ Meth)

Geng et al. 2022 DOI: 10.3109/14653249.2012.688944 8 mice/exp group (data in Mat @ Meth)

He et al. 2022 DOI: 10.1186/s40659-020-00288-1. Number of mice/groups not indicated

Power analysis was worked out and indicated that, when differences are large, as in this case, N=6 leads to clear statistical significance. Indeed, most results obtained with exon skipping have been claimed to be statistically significant but history has taught us that so far they have been clinically irrelevant, as proved by the invariable failure of phase III studies.

The researcher performing the biochemical and IF analysis received a list of codified samples that did not allow identification. But in the new mouse model, there are very few revertant fibres so that is sufficient to look at the microscope to know which is which.

2. Novelty is medium as similar approaches using exon skipping in myogenic cells have been reported previously.

Novelty is not in exon skipping itself but in allowing the snRNA to trans-correct resident nuclei of the same muscle fibre, thus amplifying of one log dystrophin production. The mdx mouse has a different mutation, so that transplanted cells would only produce their own dystrophin. This is the message of the work and is apparently missed by both Referees.

3. Medical impact is low since this approach will probably not treat DMD related dilated cardiomyopathy, the primary cause of death in DMD.

According to this general and correct statement, oligos should not be tested because they do not correct the heart, but this does not seem to be the case. AAV correct the heart but so far toxicity has largely overwhelmed efficacy. We have a MRC funded research project to convert mesoangioblasts to cardiomyocytes and transplant them when still expressing a pericyte phenotype. After appropriate controls, this strategy may be part of the final protocol.

4. The model system is adequate and the development of a humanized DMD mouse is a strength of the manuscript

We appreciate this positive comment.

Referee #3 (Remarks for Author):

This manuscript explores the use of a cell-mediated exon skipping strategy to restore dystrophin expression in myogenic cells and a humanized mouse model of Duchenne muscular dystrophy (DMD). The authors report that co-culture of corrected dystrophin deficient cells with dystrophin negative cells increased dystrophin production. They report that this occurs through transmission of exon skipping from corrected nuclei to nearby nuclei in muscle. They conclude this form of cell-mediated exon skipping may be usually in the treatment of DMD. This could be a potentially an interesting manuscript, but there are several major concerns that need to be addressed.

1. In vivo experiments of 6 animals per treatment group is low and could results in type I and type II statistical errors. Power analysis should be used to determine experimental animal numbers for the study, and this reported within the manuscript.

Please see reply to comment 1 in the above section.

2. Were those scoring experimental outcomes blinded to the treatment groups and how was genetic drift managed in the mouse colonies?

For blinding see above. We did not detect an increase in revertant fibres in the colony after more two years since Jackson lab created the colony.

3. Fig 1D: Why is dystrophin (red) localized within the cytoplasm of the muscle cells and not at the sarcolemma?

This is an in vitro IF on cells. So, the image shows a whole myotube and not only a section of it, which explains why dystrophin is apparently everywhere This is what is invariably observed with myotubes in culture. See following examples:

Meng et al. 2016 DOI: 10.1038/srep19750;

Meng et al. 2020 doi: 10.3390/ijms21197168.

<https://www.abcam.com/reagents/introducing-human-ipsc-derived-skeletal-muscle-cells-for-research-and-drug-discovery>

4. Fig 2A: There is considerable red background staining, so it is difficult to conclude if intracellular diffusion of the snRNA from a GFP positive nucleus to an adjacent nucleus occurs. Several closer nuclei to the GFP positive nucleus are completely negative so it may be helpful to quantify GFP/snRNA positive vs non-GFP/snRNA positive nuclei.

Respectfully the red fluorescence in the figure is clearly punctate and the background is low except than around the nuclei, both for increased density an overlapping DAPI staining. The red spots (snRNA) are detected everywhere, whereas only few nuclei are GFP positive since the GFP RNA is traduced around each expressing nucleus and does not diffuse as it does not bind to additional nuclear proteins. That said, this is a useful suggestion and we may add these data.

5. Fig 2B-D: GAPDH levels change in dystrophic muscle compared to unaffected muscle. In Fig 2 GAPDH levels are highly variable between the lanes.

Extraction of high molecular weight proteins from muscle tissue is a procedure affected by many variables, especially fibrosis in dystrophic muscle, that can affect solubility of dystrophin. For example high detergent concentrations are needed to solubilise dystrophin which may affect protein quantification. To compensate for possible quantification errors, we used a 2 level of normalization: first, we normalize MyHC protein expression on GAPDH (to evaluate the level of differentiation both in vivo than in vitro) and then we normalize the protein expression of Dys on the MyHC/GAPDH normalization. Therefore, even if the level of GAPDH appears at different levels in different lanes, the amount on dystrophin is anyway expressed in relation to the level of GAPDH and MyHC.

6. There is a concern that Fig 2D that GAPDH band is overexposed. The current accepted approach is to normalize dystrophin to total proteins in the same lane rather than a single housekeeping protein.

As stated above we normalize against due different proteins. Total protein normalization is a recent method that appears to provide higher reproducibility, but is not essential when differences in expression are so large as in our case.

7. Fig 2B-D: Western blots using anti-dystrophin antibodies against the N-terminus, mid-region and C-terminus of the dystrophin protein would be helpful in assessing full length vs truncated dystrophin.

This approach is useful for detecting micro-dystrophin, missing the whole central domain. In our case, exon skipping induces skipping of 4 specific exons (48-49-50-51). Thus, a discriminating antibody should only recognize an epitope in the specific skipped area. At the moment we are not aware of commercially available antibodies specific for this domain.

8. Fig 3: The dystrophin expression is not uniform, but patchy within the muscle. The authors should comment on how regional differences in dystrophin expression using this technique could affect therapeutic potential.

Intra-muscular injection causes cell distribution only in the patchy area corresponding to the needle track, as myogenic cells do not migrate for long distance in the host muscle. This is shown using Lam A/C in Fig 3C.

The

same does not occur with intra-arterial injection, that we could not perform on these mice, given their strong immune deficiency and consequent susceptibility to post-surgery infections as well as to problems related to intra-arterial injection of human cells in mice. In an intra-specific context, we may perform multiple intra-arterial injections as we did in patients, which combined with the trans-correction mechanism, should lead to dystrophin production in the therapeutic range also for patients.

9. Fig 4D: The western panel for dystrophin is highly overexposed compared to the control panels. This suggests very low levels of dystrophin are achieved and the conclusion that levels are the same at 1 month after injection based on these blots are a concern.

The overexposure was probably due to a lower binding of the secondary antibody to all samples in the blot, where lane n. 1 shows the amount of dystrophin in the TA muscle of a healthy wild type mouse. Therefore the expression of dystrophin in transplanted dystrophic mice should be compared with n. 1 and it is obvious that after one year, not one month, dystrophin persist in the muscle transplanted with U7 corrected cells much more than in muscles transplanted with wt cells.

10. Motility assays are in Fig 4C (not Fig 3C as in the manuscript text)

We thank the Referee for the comment. We will change the text accordingly.

11. The authors should use the ex vivo muscle contractile to assess muscle force generation.

The treadmill is a standard method: we retrieved 125 articles in PubMed using “treadmill” and “muscular dystrophy”, the last of April 2023. For the first time in 25 years, we saw treated mice running as long as wt mice. In contrast force analysis on isolated muscles suffers of lack of vascularization so that force decreases with time out of the body introducing a variable in the analysis (Lindsten 2016 DOI: 10.1242/jeb.124297).

12. DMD patients die prematurely from respiratory failure and dilated cardiomyopathy. The authors should comment on how this approach can be used to treated cardiopulmonary dysfunction.

We have an MRC funded research project to convert mesoangioblasts to cardiomyocytes and transplant them when still expressing a pericyte phenotype. After appropriate controls, this strategy may be part of the final protocol. At the moment it was premature to add comment about how to use this approach to treat cardiopulmonary dysfunction.

10th May 2023

Dear Dr. Galli,

Thank you for your response to the editorial decision on your manuscript entitled "Cell-mediated exon skipping normalizes dystrophin expression and muscle function in Duchenne Muscular Dystrophy". I have now carefully examined the arguments provided in your letter and discussed them with the other members of our editorial team. Additionally, I have sought external advice on the study from an expert in the field.

I am pleased to inform you that we decided to re-consider our initial decision and to invite major revision of your manuscript. Our advisor recommended a major revision and suggested that "the authors should at least repeat the in vivo experiment and confirm elevated dystrophin expression (even with low numbers of transplanted cells) and phenotypic correction in independent cohorts with all the respective controls. Inclusion of intra-arterial administration is also highly recommended to fully exploit the extravasating properties of mesoangioblasts towards body-wide correction of muscle dysfunction." Further he/she elaborated on following points:

- Regarding the issue of the dystrophin expression pattern. The main advantage of using mesoangioblasts that are able to extravasate to potentially achieve body-wide correction as opposed to the restricted impact of localized intramuscular injection was not exploited in the current study. The authors claimed that intra-arterial injections could not be performed on these mice, given their strong immune deficiency and consequent susceptibility to post-surgery infections as well as to problems related to intra-arterial injection of human cells in mice. This is not entirely convincing particularly since the authors claimed that multiple intra-arterial injections could be performed as they did in patients, which combined with the trans-correction mechanism, "should lead to dystrophin production in the therapeutic range also for patients". Inclusion of a cohort based on intra-arterial administration of corrected mesoangioblasts would significantly strengthen the manuscript.
- The statistical significance should be indicated more clearly and directly in the context of each dataset (in legends and results section) rather than just in the materials and methods since it was not immediately clear how often the experiments were repeated and based on how many replicates. In particular, there is still the outstanding question as to how often the key in vivo experiments were repeated. Even if n=6 mice were used per cohort, as indicated in the materials and methods, confirming the dystrophin expression and motility data in a repeated in vivo experiment would certainly strengthen the paper and increase the reviewers' confidence in the results and conclusions. Simply implying that the experiment would not need to be repeated because of animal ethics considerations (the so-called 3R principle) is not entirely convincing since it is important to exclude "systematic" statistical errors (type I and II).
- There is some "sloppiness" in the manuscript and figure organization and I would also suggest subduing some of the claims to avoid "over-selling" the research findings, especially given the outstanding questions. However, this is a matter of style over substance and could relatively easily be addressed without conducting additional experiments.

Please provide detailed responses to the referee concerns using only scientific argumentation and appropriately amend the manuscript for clarity and to strengthen the main message of the study. Criticism regarding cardiopulmonary corrections should be addressed by discussion in point-by-point response and in the manuscript text.

We would welcome the submission of a revised version within three to six months for further consideration. Please let us know if you require longer to complete the revision.

I look forward to receiving your revised manuscript.

Yours sincerely,

Zeljko Durdevic

Zeljko Durdevic

We require:

- 1) A .docx formatted version of the manuscript text (including legends for main figures, EV figures and tables). Please make sure that the changes are highlighted to be clearly visible.
 - 2) Individual production quality figure files as .eps, .tif, .jpg (one file per figure). For guidance, download the 'Figure Guide PDF': (<https://www.embopress.org/page/journal/17574684/authorguide#figureformat>).
 - 3) A .docx formatted letter INCLUDING the reviewers' reports and your detailed point-by-point responses to their comments. As part of the EMBO Press transparent editorial process, the point-by-point response is part of the Review Process File (RPF), which will be published alongside your paper.
 - 4) A complete author checklist, which you can download from our author guidelines (<https://www.embopress.org/page/journal/17574684/authorguide#submissionofrevisions>). Please insert information in the checklist that is also reflected in the manuscript. The completed author checklist will also be part of the RPF.
 - 5) Please note that all corresponding authors are required to supply an ORCID ID for their name upon submission of a revised manuscript.
 - 6) It is mandatory to include a 'Data Availability' section after the Materials and Methods. Before submitting your revision, primary datasets produced in this study need to be deposited in an appropriate public database, and the accession numbers and database listed under 'Data Availability'. Please remember to provide a reviewer password if the datasets are not yet public (see <https://www.embopress.org/page/journal/17574684/authorguide#dataavailability>).
- In case you have no data that requires deposition in a public database, please state so in this section. Note that the Data Availability Section is restricted to new primary data that are part of this study.
- 7) For data quantification: please specify the name of the statistical test used to generate error bars and P values, the number (n) of independent experiments (specify technical or biological replicates) underlying each data point and the test used to calculate p-values in each figure legend. The figure legends should contain a basic description of n, P and the test applied. Graphs must include a description of the bars and the error bars (s.d., s.e.m.). See also 'Figure Legend' guidelines: <https://www.embopress.org/page/journal/17574684/authorguide#figureformat>
 - 8) At EMBO Press we ask authors to provide source data for the main manuscript figures. Our source data coordinator will contact you to discuss which figure panels we would need source data for and will also provide you with helpful tips on how to upload and organize the files.
 - 9) Our journal encourages inclusion of *data citations in the reference list* to directly cite datasets that were re-used and obtained from public databases. Data citations in the article text are distinct from normal bibliographical citations and should directly link to the database records from which the data can be accessed. In the main text, data citations are formatted as follows: "Data ref: Smith et al, 2001" or "Data ref: NCBI Sequence Read Archive PRJNA342805, 2017". In the Reference list, data citations must be labeled with "[DATASET]". A data reference must provide the database name, accession number/identifiers and a resolvable link to the landing page from which the data can be accessed at the end of the reference. Further instructions are available at .
 - 10) We replaced Supplementary Information with Expanded View (EV) Figures and Tables that are collapsible/expandable online. A maximum of 5 EV Figures can be typeset. EV Figures should be cited as 'Figure EV1, Figure EV2' etc... in the text and their respective legends should be included in the main text after the legends of regular figures.

- Additional Tables/Datasets should be labeled and referred to as Table EV1, Dataset EV1, etc. Legends have to be provided in

a separate tab in case of .xls files. Alternatively, the legend can be supplied as a separate text file (README) and zipped together with the Table/Dataset file.

13) Author contributions: You will be asked to provide CRediT (Contributor Role Taxonomy) terms in the submission system. These replace a narrative author contribution section in the manuscript.

14) A Conflict of Interest statement should be provided in the main text.

Please note: When submitting your revision you will be prompted to enter your funding and payment information. This will allow Wiley to send you a quote for the article processing charge (APC) in case of acceptance. This quote takes into account any reduction or fee waivers that you may be eligible for. Authors do not need to pay any fees before their manuscript is accepted and transferred to the publisher.

EMBO Press participates in many Publish and Read agreements that allow authors to publish Open Access with reduced/no publication charges. Check your eligibility: <https://authorservices.wiley.com/author-resources/Journal-Authors/open-access/affiliation-policies-payments/index.html>

Referee #1 (Comments on Novelty/Model System for Author):

1- Quantification for many experiments is missing, and in some cases, it appears the graph represents only one sample. The "n" per group is not clearly described (one needs to search in M&M). This information should be provided in Figure Legends.

All the WB are quantified so that this first criticism lacks substance. Only in one experiment Fig 2C, we have only one sample because we had access to a muscle biopsy of only one DMD patient with the specific mutation needed. We apologise for not having indicated "n" in the Figure legend and will do this, but it is very different from non-indicating the replicates at all.

2- Mesoangioblasts are not novel and the the lentiviral vector expressing the U7 snRNA was first described 20 years ago, as mentioned in the introduction (De Angelis et al., PNAS, 2002).

In this comment the Referee misses the point that the use of a cell mediated exon-skipping brings dystrophin production to an unprecedented therapeutic level (other than with AAV) and this is novel and potentially ground-breaking

3- There is potential for impact but considering all the issues described in detail below, for now, medical impact is medium.

This is the Referee opinion, probably because she/he disregarded the novel mechanism and the level of dystrophin expression, not the least relevant point for DMD therapy.

Referee #1 (Remarks for Author):

Francesco Galli, Giulio Cossu and colleagues report a cell-mediated exon skipping protocol to treat Duchenne Muscular Dystrophy (DMD). Given the limited engraftment encountered in recent clinical trials assessing transplantation of mesoangioblasts in DMD patients, they postulate that cell therapy wasn't (and won't) be effective in DMD due to the fibrotic state of DMD muscles. Based on this premise, they developed a therapy in which minimal cell engraftment may lead to some therapeutic benefit through cross-correction of resident dystrophic nuclei by the U7 small nuclear RNA engineered to skip exon 5 of the dystrophin gene. While the study is of interest, there are numerous concerns with this manuscript in its present form.

Major comments:

1- The introduction is confusing. Cossu and colleagues have published a body of literature on the regenerative potential of mesoangioblasts, which led to a first-in-human clinical trial. Because this trial resulted in limited engraftment (<1%), which could have been for several reasons, including low cell dose and age of patients at enrollment (as suggested by the authors), but also due to inefficient immunosuppression and/or lack of regenerative potential of injected cells, the authors postulate in the introduction that "even starting at an earlier age and implementing the protocol, engraftment will always remain low in tissues such as muscle and brain where ablation of diseased resident cells is not possible". There is no scientific basis for this statement, nor for the comparison between muscle and brain. The reference listed is a review by Thomas Rando's group that does not support this claim. In addition, this statement goes against all the literature published by this group in the last 15 years promising to treat muscular dystrophy with mesoangioblasts. Oddly, in the discussion, the authors make overpromising claims again, such as "this work is ground-breaking as we can now produce "therapeutic levels of dystrophin", and their intent to cure young patients.

The review by Thomas Rando (doi: 10.1172/JCI142031) refers to the sentence "Degeneration is followed by regeneration carried out by satellite cells, resident myogenic stem/progenitor cells and, to a minor extent, interstitial cells such as pericytes" and not to what the reviewer states. The reference listed for the "even starting at an earlier age and implementing the protocol, engraftment will always remain low in tissues such as muscle and brain where ablation of diseased resident cells is not possible" is (Cossu G. et al. doi: 10.1016/S0140-6736(17)31366-1). This is a commonly accepted concept in the field and dates to the first years of bone marrow transplantation when myeloablation increased engraftment leading to successful outcomes. It is taught in medical schools.

The last sentence is wrong and misleading. At variance with Companies that continue, trial after trial, to claim efficacy and safety with nothing changing in the progress of the disease, we never promised any therapy; we had encouraging results in animals (Sampaolesi et al. 2003, doi:10.1126/science.1082254; Sampaolesi et al. 2006, doi:10.1038/nature05282), but with multiple, intra-specific or even syngeneic transplantations; based on these results we run a phase I/IIa clinical trial and reported lack of efficacy (Cossu et al. 2015 doi: 10.15252/emmm.201505636.), even though the amount of dystrophin we detected was more than what currently mentioned as "efficacious" in a recent report (Servais et al. 2022, doi: 10.1089/nat.2021.0043.) with no western blot shown (also because the amount reported, 1.09% of a healthy muscle, would not be detectable on WB). Indeed, Referee fails to evaluate the amount of dystrophin and the functional recovery, that are the only novel results that justify a cautious optimism.

2- Key information is missing. The authors mentioned the peak of degeneration in this novel mouse model is at one month of age but there is no information about the timing of transplantation. The levels of fibrosis are also not

clear. This is critical given the authors claim that poor engraftment was due to advanced fibrotic pathology. The authors also do not explain whether the mice were subjected to any type of pre-injury prior to transplantation - was cardiotoxin or other injury performed? What are the levels of engraftment? It is necessary to show clear quantification of engraftment (IF and western blot). Are engraftment levels lower in this mouse model, opposed to previous studies involving the transplantation of mesoangioblasts? Are engraftment levels low because cells were injected in fibrotic muscle? Differences and commonalities between previous and current studies should be acknowledged.

We apologise for the missing information on the timing of the transplantation (about 6 months of age); it will be added in a revised version. The phenotype of this mouse is similar to the classic mdx mouse, and the level of fibrosis (Suppl. Fig 3C) is comparable to what reported in mdx.

We do not explain experiments that we did not do: we did not induce injury, e.g., with CTX as it confuses original pathology with acute damage and would never be allowed in a patient. The quantification of engraftment was carried out by IF (shown in Fig.3C) using Lam A/C, a specific human marker and is clearly shown in the graph. Differences with previous papers were due to different protocols (multiple injections, not performed here, use of WT mouse cells in mice, and lack of quantification of engraftment in papers published 20 years ago).

3- Overall, figure organization and their respective descriptions in the text is chaotic. For instance, not all figures are described, and in many instances, they are covered out of order (for example, Fig 1C is described before 1A).

Fig 1C is described before Fig 1A to indicate the level of transduction of the cells, important to explain the results in FIG 1A and the correct percentage of the ratio of transduced cells for the results in Fig 1D. We rearranged the text to follow the order of the figures.

4- Figure 2B: It is important to demonstrate whether there are differences in engraftment between WT/DMD and DMDu7/DMD. For some samples, there are no error bars suggesting that they quantified only one gel. Protein expression results are more relevant than the RNA data shown in Figure 1. Better attention to rigor is required

Figure 2B is an in vitro experiment and so there is no engraftment to be quantified. A comparison of the cells' ratio between WT/DMD and DMDU7/DMD is shown in figure 1C. All the experiment were made in triplicate, and we will add the error bar to Fig 2B. We agree about the relevance of protein expression, but RNA data, shown in figure 1A, only show the ability of snRNA to induce the exon skipping and to generate the correct shorter RNA sequence, not to quantify the amount of dystrophin mRNA produced.

5- Figure S3A: In addition to merged images, the authors should include images for laminin and Dys staining separately as it is not easy to visualize Dys staining in the merge. The TA staining looks the same between WT and mdx51 mice. No information is provided regarding the age of the mice analyzed in this figure. The authors should provide images with better resolution and higher magnification.

We provided separate images for a better visualization, and we will add information about the age of the animal used. The choice of the magnification is to show a large area of the muscle and not only a selected area, and so have a broader view of the tissue section.

6- Fig 3A and 3B: The authors quantify the western blot but find no significant difference between DMD and WT. What does this mean? It is claimed that DMD u7 cells cross-correct resident dystrophic nuclei to produce greater quantities of dystrophin, but the data seems to be saying something different.

It means that the U7-mediated gene correction leads to unprecedented accumulation of dystrophin, and this is confirmed by the much lower amount of dystrophin produced by muscles transplanted with the same number of WT cells. It approaches 50% of the level of the WT mouse. We also added the statistically significant difference between WT and U7 like requested by the reviewer.

7- It is difficult to find the number of mice used in different experiments. This information should be included in each figure legend.

We provided the information also in the figure legend.

8- Fig 4: The authors should include a negative control for the sn U7 staining so the reader can appreciate what the background level is.

A negative control for the snU7 is shown in Fig S2, where part of the cells was transduced with a control LV expressing only GFP. No dystrophin was detected. We disagree on the relevance of this control for the in vivo experiments. A negative control for the snU7 won't give any information about background level for the simple fact that antibodies do not recognize the snRNA but the product of the exon-skipping, dystrophin protein that we show by IF and WB.

Minor comments:

- Results describing cell plug as described in Fig. S5, not Fig. S4 as described in the text.

We corrected the error in the text.

Referee #3 (Comments on Novelty/Model System for Author):

1. The in vivo experiments were on just 6 animals per treatment group which is low and could result in type I and type II statistical errors. Power analysis was not reported in the study. There are concerns with experimental rigor including those scoring experimental outcomes blinded to the treatment groups. *As indicated in the paragraph on statistical analysis, the number of 6 animals in in vivo experiments represents the best compromise between the 3R guidelines and the need to have reproducibility and consistency in the results.*

*We checked recently published papers on the topic and report the first three papers we found:
Sun et al. 2022 DOI: 10.1016/j.stem.2022.03.004: 6 mice/exp group (data in Mat @ Meth)
Geng et al. 2022 DOI: 10.3109/14653249.2012.688944 8 mice/exp group (data in Mat @ Meth)
He et al. 2022 DOI: 10.1186/s40659-020-00288-1. Number of mice/groups not indicated.*

Power analysis was worked out using previous experiments. We used Z-score 1.96 for 95% confidence, population standard deviation 0.5, effect size 0.5, indicating we need a sample size of approximately 4 to achieve a clear statistical significance. In our case N=6 leads to clear statistical significance. Indeed, most results obtained with exon skipping have been claimed to be statistically significant, but history has taught us that so far, they have been clinically irrelevant, as proved by the invariable failure of phase III studies.

The researcher performing the biochemical and IF analysis received a list of codified samples that did not allow identification. But in the new mouse model, there are very few revertant fibres so that is sufficient to look at the microscope to know which is which.

2. Novelty is medium as similar approaches using exon skipping in myogenic cells have been reported previously

Novelty is not in exon skipping itself but in allowing the snRNA to trans-correct resident nuclei of the same muscle fibre, thus amplifying of one log dystrophin production. The mdx mouse has a different mutation, so that transplanted cells would only produce their own dystrophin. This is the message of the work and is apparently missed by both Referees.

3. Medical impact is low since this approach will probably not treat DMD related dilated cardiomyopathy, the primary cause of death in DMD

According to this general and correct statement, oligos should not be tested because they do not correct the heart, but this does not seem to be the case. AAV correct the heart but so far toxicity has largely overwhelmed efficacy. We have a MRC funded research project to convert mesoangioblasts to cardiomyocytes and transplant them when still expressing a pericyte phenotype. After appropriate controls, this strategy may be part of the final protocol. We add a sentence to explain this in the text.

4. The model system is adequate and the development of a humanized DMD mouse is a strength of the manuscript

We appreciate this positive comment.

Referee #3 (Remarks for Author):

This manuscript explores the use of a cell-mediated exon skipping strategy to restore dystrophin expression in myogenic cells and a humanized mouse model of Duchenne muscular dystrophy (DMD). The authors report that co-culture of corrected dystrophin deficient cells with dystrophin negative cells increased dystrophin production. They report that this occurs through transmission of exon skipping from corrected nuclei to nearby nuclei in muscle. They conclude this form of cell-mediated exon skipping may be usually in the treatment of DMD. This could be a potentially an interesting manuscript, but there are several major concerns that need to be addressed.

1. In vivo experiments of 6 animals per treatment group is low and could result in type I and type II statistical errors. Power analysis should be used to determine experimental animal numbers for the study, and this reported within the manuscript.

Please see reply to comment 1 in the above section.

2. Were those scoring experimental outcomes blinded to the treatment groups and how was genetic drift managed in the mouse colonies?

For blinding see above. We did not detect an increase in revertant fibres in the colony after more two years since Jackson lab created the colony.

3. Fig 1D: Why is dystrophin (red) localized within the cytoplasm of the muscle cells and not at the sarcolemma?

This is an in vitro IF on cells. So, the image shows a whole myotube and not only a section of it, which explains why dystrophin is apparently everywhere This is what is invariably observed with myotubes in culture. See following examples:

Meng et al. 2016 DOI: 10.1038/srep19750;

Meng et al. 2020doi: 10.3390/ijms21197168.

<https://www.abcam.com/reagents/introducing-human-ipsc-derived-skeletal-muscle-cells-for-research-and-drug-discovery>

4. Fig 2A: There is considerable red background staining, so it is difficult to conclude if intracellular diffusion of the snRNA from a GFP positive nucleus to an adjacent nucleus occurs. Several closer nuclei to the GFP positive nucleus are completely negative so it may be helpful to quantify GFP/snRNA positive vs non-GFP/snRNA positive nuclei.

Respectfully the red fluorescence in the figure is clearly punctate and the background is low except than around the nuclei, both for increased density an overlapping DAPI staining. The red spots (snRNA) are detected everywhere, whereas only few nuclei are GFP positive since the GFP RNA is traduced around each expressing nucleus and does not diffuse as it does not bind to additional nuclear proteins.

5. Fig 2B-D: GAPDH levels change in dystrophic muscle compared to unaffected muscle. In Fig 2 GAPDH levels are highly variable between the lanes.

Extraction of high molecular weight proteins from muscle tissue is a procedure affected by many variables, especially fibrosis in dystrophic muscle, that can affect solubility of dystrophin. For example high detergent concentrations are needed to solubilise dystrophin which may affect protein quantification. To compensate for possible quantification errors, we used a 2 level of normalization: first, we normalize MyHC protein expression on GAPDH (to evaluate the level of differentiation both in vivo than in vitro) and then we normalize the protein expression of Dys on the MyHC/GAPDH normalization. Therefore, even if the level of GAPDH appears at different levels in different lanes, the amount on dystrophin is anyway expressed in relation to the level of GAPDH and MyHC.

6. There is a concern that Fig 2D that GAPDH band is overexposed. The current accepted approach is to normalize dystrophin to total proteins in the same lane rather than a single housekeeping protein.

As stated above we normalized against two different proteins. Total protein normalization is a recent method that appears to provide higher reproducibility but is not essential when differences in expression are so large as in our case.

7. Fig 2B-D: Western blots using anti-dystrophin antibodies against the N-terminus, mid-region and C-terminus of the dystrophin protein would be helpful in assessing full length vs truncated dystrophin.

This approach is useful for detecting micro-dystrophin, missing the whole central domain. In our case, exon skipping induces skipping of 4 specific exons (48-49-50-51). Thus, a discriminating antibody should only recognize an epitope in the specific skipped area. Currently we are not aware of commercially available antibodies specific for this domain.

8. Fig 3: The dystrophin expression is not uniform, but patchy within the muscle. The authors should comment on how regional differences in dystrophin expression using this technique could affect therapeutic potential.

Intra-muscular injection causes cell distribution only in the patchy area corresponding to the needle track, as myogenic cells do not migrate for long distance in the host muscle. This is shown using Lam A/C in Fig 3C.

9. Fig 4D: The western panel for dystrophin is highly overexposed compared to the control panels. This suggests very low levels of dystrophin are achieved and the conclusion that levels are the same at 1 month after injection based on these blots are a concern.

*The overexposure was probably due to a lower binding of the secondary antibody to all samples in the blot, where lane n. 1 shows the amount of dystrophin in the TA muscle of a healthy wild type mouse. Therefore, the expression of dystrophin in transplanted dystrophic mice should be compared with n. 1 and it is obvious that **after eleven, not one month**, dystrophin persists in the muscle transplanted wit U7 corrected cells much more than in muscles transplanted with WT cells.*

10. Motility assays are in Fig 4C (not Fig 3C as in the manuscript text)

We thank the Referee for the comment. We changed the text accordingly.

11. The authors should use the ex vivo muscle contractile to assess muscle force generation.

The treadmill is a standard method: we retrieved over 100 articles in PubMed using "treadmill" and "muscular dystrophy", the last of April 2023. For the first time in 25 years, we saw treated mice running as long as WT mice. In contrast force analysis on isolated muscles suffers of lack of vascularization so that force decreases with time out of the body introducing a variable in the analysis (Lindsten 2016 DOI: 10.1242/jeb.124297).

12. DMD patients die prematurely from respiratory failure and dilated cardiomyopathy. The authors should comment on how this approach can be used to treated cardiopulmonary dysfunction.

We have an MRC funded research project to convert mesoangioblasts to cardiomyocytes and transplant them when still expressing a pericyte phenotype. After appropriate controls, this strategy may be part of the final protocol. We explained this in the text.

Extra Referee

- Regarding the issue of the dystrophin expression pattern. The main advantage of using mesoangioblasts that are able to extravasate to potentially achieve body-wide correction as opposed to the restricted impact of localized intramuscular injection was not exploited in the current study. The authors claimed that intra-arterial injections could not be performed on these mice, given their strong immune deficiency and consequent susceptibility to post-surgery infections as well as to problems related to intra-arterial injection of human cells in mice. This is not entirely convincing particularly since the authors claimed that multiple intra-arterial injections could be performed as they did in patients, which combined with the trans-correction mechanism, "should lead to dystrophin production in the therapeutic range also for patients". Inclusion of a cohort based on intra-arterial administration of corrected mesoangioblasts would significantly strengthen the manuscript.

We agree that this is an important part of the work, despite the risk related to extreme immune deficiency (NSG vs SCID or WT of previous work), the lower efficiency of xenogeneic transplantation and the limited numbers of animals of the right age available at the time of the review. Therefore we conducted one experiment based upon a single intra-arterial injection, and we added the results of the experiment in the manuscript (Figure 5). The rationale for a single injection is based on two criteria: 1. By reducing the number of cells injected we would better appreciate the difference between genetically corrected and WT cells, also in the perspective of reduced efficacy moving from mice to patients; 2. We would reduce the risk of infections with repeated injections. Indeed we used 8 animals but 3 of them died for post-surgery infections. For this reason, we could perform analysis only on 5 animals, and in all cases we could detect a dystrophin band, though faint, in animals transplanted with genetically corrected cells but not in those transplanted with WT cells. Indeed in all previous experiments we needed three to five consecutive injections to clearly detect dystrophin by WB.

- The statistical significance should be indicated more clearly and directly in the context of each dataset (in legends and results section) rather than just in the materials and methods since it was not immediately clear how often the experiments were repeated and based on how many replicates. In particular, there is still the outstanding question as to how often the key in vivo experiments were repeated. Even if n=6 mice were used per cohort, as indicated in the materials and methods, confirming the dystrophin expression and motility data in a repeated in vivo experiment would certainly strengthen the paper and increase the reviewers' confidence in the results and conclusions. Simply implying that the experiment would not need to be repeated because of animal ethics considerations (the so-called 3R principle) is not entirely convincing since it is important to exclude "systematic" statistical errors (type I and II)."

We thank the Referee for the comment. We changed the text accordingly indicating more clearly the number of animals used.

While repeating experiments enhances confidence on data reproducibility, we would like to point out that for in vitro experiments we used three different myogenic cell types of Figures 2 B-D), all indicating that genetically corrected DMD cells produce much more dystrophin than WT cells (from 3 to 10 fold more) when cocultured with an excess of dystrophic myoblasts. For in vivo experiments we report many different experiments: in vivo transplantation with two different cell types (Figure 3 A, B), a dilution experiment (Figure 4B), a time course experiment (Figure 4D), the newly added intra-arterial transplantation (Figure 5A) and the Matrigel plug (Supplemental Figure 5B). All the experiments concur that genetically corrected DMD myogenic cells produce much more dystrophin than WT cells in vivo. The systemic statistical error is the expected value of the overall error. It is a different problem than random error that cannot be reduced by simple repetition either of the experiment or of the measurement procedure but can be minimised by combination of different experiments and alternative measurement procedures (<https://www.statistics.com/glossary/systematic-error/#:~:text=Statistical%20Glossary&text=Systematic%20error%20is%20the%20error.value%20of%20the%20overall%20error>). It is also well established that power could be adequate with a smaller number of samples if the observed effect is large, like we observed in our case where the P value is <0.001 (<https://www.ncbi.nlm.nih.gov/books/NBK557530/#:~:text=A%20type%20I%20error%20occurs,there%20indeed%20was%20no%20difference>).

We also added information about the power analysis used in the statistical analysis paragraph.

- There is some "sloppiness" in the manuscript and figure organization, and I would also suggest subduing some of the claims to avoid "over-selling" the research findings, especially given the outstanding questions. However,

this is a matter of style over substance and could relatively easily be addressed without conducting additional experiments.

We thank the Referee for the comment. We changed the text accordingly.

Please provide detailed responses to the referee concerns using only scientific argumentation and appropriately amend the manuscript for clarity and to strengthen the main message of the study. Criticism regarding cardiopulmonary corrections should be addressed by discussion in point-by-point response and in the manuscript text.

We addressed the criticism concerning the cardiopulmonary correction in the discussion. At the moment we considered premature, having only preliminary data, to describe in detail the use this approach to treat cardiopulmonary dysfunction. However, we have an MRC funded research project to convert mesoangioblasts to cardiomyocytes and transplant them when still expressing a pericyte phenotype.

21st Dec 2023

Dear Dr. Galli,

Thank you for the submission of your revised manuscript to EMBO Molecular Medicine. We have now received feedback from the two reviewers who agreed to evaluate your manuscript. As you will see from their reports pasted below, while referee #4 supports publication of the manuscript, referee #3 acknowledges the improvements of the revised manuscript but also raises several concerns, particularly regarding the low number of animals per experiment. Based on the referee reports and after an editorial discussion, we agreed that the authors responded adequately to the referees' criticism. Therefore, I am pleased to inform you that we will be able to accept your manuscript pending the following final amendments:

1) Please address all referee #3 concerns. Points 1, 3, and 4 should be discussed and where appropriate acknowledge limitations of the study, for point #2 please provide and quantify less exposed western blots.

2) Figures:

- Main figures and EV figures should be uploaded as individual high-resolution files and only the legends should be in the manuscript text. Please rename suppl. figures to "Figure EV1" etc. and upload as individual high-resolution figure files. Legends should be in the manuscript, after the main figure legends, with the heading "Expanded View Figure Legends."

- During a standard image analysis we detected potential aberrations in the figure set, and we would like to clarify these issues before accepting your manuscript for publication. We kindly invite you to check images in Figure S2A and B and explain the absence of signal in GFP, Dys and DAPI/GFP/Dys (no virus) in S2A and Dys (T2A GFP) S2B. We kindly ask you to check this and to provide source data for these figures.

3) In the main manuscript file, please do the following:

- Please address all comments suggested by our data editors listed below:

o DAS: Please note that the data availability statement is not provided in the manuscript.

o Figure legends:

1. Please note that the figure legend style does not comply with the journal guidelines i.e. all the figure legends are in a run-on style

2. Please note that a separate 'Data Information' section is required in the legends of figures 2; 3; 4; 5.

3. Please indicate the statistical test used for data analysis in the legends of figures 1e; 3a-c; 4b-d; supplementary figure 4.

4. Although 'n' is provided, please describe the nature of entity for 'n' in the legends of figures 1e; 2b-d; 3a-c; 4b-d; 5a; supplementary figure 4.

5. Please note that the error bars are not defined in the legends of figures 1e; 2b-d; 3a-b; 4b-d; 5a; supplementary figure 4.

6. Please note that scale bar and its definition are missing for supplementary figures 3a-c.

- Add up to 5 keywords.

- Add callouts for Fig 5B. All figures should be called out in a sequential order. Currently, Fig S5 is called out before Fig S4. Please correct.

- Please rename "Conflict of Interest" to "Disclosure Statement & Competing Interests" and place it after "Acknowledgements" below the M&M section. Also, please add the following sentence: Giulio Cossu is an Editorial Advisory Board Member. This has no bearing on the editorial consideration of this article for publication. We updated our journal's competing interests policy in January 2022 and request authors to consider both actual and perceived competing interests. Please review the policy <https://www.embopress.org/competing-interests> and update your competing interests if necessary.

- Please add "Data availability" and place it after M&M section. If no data are deposited in public repositories add the sentence: This study includes no data deposited in external repositories.

- Please correct the reference citation in the text and reference list. In the text of the manuscript, a reference should be cited by author and year of publication. Include a space between a word and the opening parenthesis of the reference that follows. In the reference list, citations should be listed in alphabetical order. Where there are more than 10 authors on a paper, 10 will be listed, followed by "et al.". Please check "Author Guidelines" for more information.

<https://www.embopress.org/page/journal/17574684/authorguide#referencesformat>

4) The Paper Explained: Please provide "The Paper Explained" and add it to the main manuscript text. Please check "Author Guidelines" for more information. <https://www.embopress.org/page/journal/17574684/authorguide#researcharticleguide>

5) Synopsis: Every published paper now includes a 'Synopsis' to further enhance discoverability. Synopses are displayed on the journal webpage and are freely accessible to all readers. They include separate synopsis image and synopsis text.

- Synopsis image: Please provide the visual abstract as a high-resolution jpeg file 550 px-wide x (250-400)-px high.

- Synopsis text: Please provide a short standfirst (maximum of 300 characters, including space) as well as 2-5 one sentence bullet points that summarise the paper as a .doc file. Please write the bullet points to summarise the key NEW findings. They should be designed to be complementary to the abstract - i.e. not repeat the same text. We encourage inclusion of key acronyms and quantitative information (maximum of 30 words / bullet point). Please use the passive voice.

6) For more information: This space should be used to list relevant web links for further consultation by our readers. Could you identify some relevant ones and provide such information as well? Some examples are patient associations, relevant databases, OMIM/proteins/genes links, author's websites, etc...

7) As part of the EMBO Publications transparent editorial process initiative (see our Editorial at <http://embomolmed.embopress.org/content/2/9/329>), EMBO Molecular Medicine will publish online a Review Process File (RPF) to accompany accepted manuscripts. This file will be published in conjunction with your paper and will include the anonymous referee reports, your point-by-point response and all pertinent correspondence relating to the manuscript. Let us know whether you agree with the publication of the RPF and as here, if you want to remove or not any figures from it prior to publication. Please note that the Authors checklist will be published at the end of the RPF.

8) Please provide a point-by-point letter INCLUDING my comments as well as the reviewer's reports and your detailed responses (as Word file).

I look forward to reading a new revised version of your manuscript as soon as possible.

Yours sincerely,

Zeljko Durdevic

*** Instructions to submit your revised manuscript ***

1) a .docx formatted version of the manuscript text (including Figure legends and tables)

2) Separate figure files*

3) supplemental information as Expanded View and/or Appendix. Please carefully check the authors guidelines for formatting Expanded view and Appendix figures and tables at <https://www.embopress.org/page/journal/17574684/authorguide#expandedview>

4) a letter INCLUDING the reviewer's reports and your detailed responses to their comments (as Word file).

5) The paper explained: EMBO Molecular Medicine articles are accompanied by a summary of the articles to emphasize the major findings in the paper and their medical implications for the non-specialist reader. Please provide a draft summary of your article highlighting

6) For more information: There is space at the end of each article to list relevant web links for further consultation by our readers. Could you identify some relevant ones and provide such information as well? Some examples are patient associations, relevant databases, OMIM/proteins/genes links, author's websites, etc...

7) Author contributions: the contribution of every author must be detailed in a separate section.

8) EMBO Molecular Medicine now requires a complete author checklist (<https://www.embopress.org/page/journal/17574684/authorguide>) to be submitted with all revised manuscripts. Please use the checklist as guideline for the sort of information we need WITHIN the manuscript. The checklist should only be filled with page numbers where the information can be found. This is particularly important for animal reporting, antibody dilutions (missing) and exact values and n that should be indicated instead of a range.

9) Every published paper now includes a 'Synopsis' to further enhance discoverability. Synopses are displayed on the journal webpage and are freely accessible to all readers. They include a short stand first (maximum of 300 characters, including space) as well as 2-5 one sentence bullet points that summarise the paper. Please write the bullet points to summarise the key NEW findings. They should be designed to be complementary to the abstract - i.e. not repeat the same text. We encourage inclusion of key acronyms and quantitative information (maximum of 30 words / bullet point). Please use the passive voice. Please attach these in a separate file or send them by email, we will incorporate them accordingly.

You are also welcome to suggest a striking image or visual abstract to illustrate your article. If you do please provide a jpeg file 550 px-wide x 300-800px high.

10) A Conflict of Interest statement should be provided in the main text

11) Please note that we now mandate that all corresponding authors list an ORCID digital identifier. This takes <90 seconds to complete. We encourage all authors to supply an ORCID identifier, which will be linked to their name for unambiguous name identification.

Currently, our records indicate that the ORCID for your account is 0000-0002-6696-4086.

Link Not Available

Photos 400-800 DPI

*Additional important information regarding figures and illustrations can be found at <https://bit.ly/EMBOPressFigurePreparationGuideline>. See also figure legend preparation guidelines: <https://www.embopress.org/page/journal/17574684/authorguide#figureformat>

***** Reviewer's comments *****

Referee #3 (Comments on Novelty/Model System for Author):

Analysis using N=3 or 6 mice is low which could result in statistical error. Including more animals in the treatment groups would strengthen the study and conclusions. Exon skipping studies in cells have previously been reported so this concept is not entirely novel. Medical impact remains moderate primarily due to a lack of functional studies.

Referee #3 (Remarks for Author):

The manuscript has been improved and the authors have addressed some concerns. There are concerns that remain or were not adequately addressed.

1. Functional muscles assay are limited only to motility. Muscle ex vivo or in vivo contractility assays are the gold standard for measuring improvements in muscle strength in mouse models of DMD.

2. Western blots still appear overexposed and it is unclear if the exposures are within the linear range to accurately measure

protein levels.

3. The authors have attempted to justify the experimental animal numbers (n=3 or 6) in the study. There are still concerns over statistical errors using these low numbers.

4. The authors now report a 10% increase in dystrophin after intra-arterial injections. Is this level of dystrophin adequate to improve muscle function in mdx mice and DMD patients?

Referee #4 (Comments on Novelty/Model System for Author):

This is an important study and I felt that the paper certainly has merit in terms of novelty and potential translational implications. The main conclusion of the study is that transplantation of myogenic cells corrected by snRNA-based exon skipping can correct distal (non-transduced) nuclei in myofibers of dystrophic mice. Most importantly, this resulted in an unexpected disproportionate beneficial effect both in terms of (i) dystrophin expression levels and (ii) phenotypic correction efficiencies based on a motility assay. This study has important implications by demonstrating that clinically meaningful levels of dystrophin expression could be attained, even in the face of relatively low engraftment efficiencies of gene-modified myogenic cells by virtue of the diffusion of the exon-skipping snRNA towards non-transduced nuclei in the myofibers. Even if exon-skipping or the use of gene-corrected mesoangioblast is not novel in as of itself, the conclusions of the distal disproportionate effects on dystrophin expression and phenotypic corrections are sufficiently novel and clinically relevant. Previous experiments in published papers based on exon-skipping or mesoangioblasts were not designed to address this important question.

Referee #4 (Remarks for Author):

The author adequately addressed my comments. In particular, the inclusion of the data obtained after intra-arterial delivery significantly strengthen the manuscript.

Referee #3 (Comments on Novelty/Model System for Author):

Analysis using N=3 or 6 mice is low which could result in statistical error. Including more animals in the treatment groups would strengthen the study and conclusions. Exon skipping studies in cells have previously been reported so this concept is not entirely novel. Medical impact remains moderate primarily due to a lack of functional studies.

We used 6 mice for most experimental group. The numerosity depends on the extent of the differences observed. When differences are small, more animals are needed, but in our case differences were huge.

The concept is not novel, the combination of exon skipping, and cell therapy is novel.

We have treadmill experiments, that for the first time in our twenty-five years' experience and for what reported in the literature, equals the motility of WT mice.

Referee #3 (Remarks for Author):

The manuscript has been improved and the authors have addressed some concerns. There are concerns that remain or were not adequately addressed.

1-Functional muscles assay are limited only to motility. Muscle ex vivo or in vivo contractility assays are the gold standard for measuring improvements in muscle strength in mouse models of DMD.

Muscles ex vivo are agonizing cells in hypoxic conditions. In vivo contractility is one method to measure muscle force, but in the end the animal motility is the most comprehensive functional measure, just like the 6 minutes-walk tests in patients.

2-Western blots still appear overexposed and it is unclear if the exposures are within the linear range to accurately measure protein levels.

The criticism does not consider the striking differences among the different dystrophin bands. Overexposure tends to mask differences by oversaturating the signal so that they can be missed. However, differences are still very clear in this specific case.

3-The authors have attempted to justify the experimental animal numbers (n=3 or 6) in the study. There are still concerns over statistical errors using these low numbers.

We used 6 mice for most experimental group. The numerosity depends on the extent of the differences observed. When differences are small, more animals are needed, but in our case differences were huge.

4-The authors now report a 10% increase in dystrophin after intra-arterial injections. Is this level of dystrophin adequate to improve muscle function in mdx mice and DMD patients?

A 10% increase in dystrophin would be inadequate to improve muscle function. More than twenty years ago we published that to get a significant amelioration in dystrophic mice, three consecutive intra-arterial injections were needed using WT cells (Sampaolesi et al. Science 2003). In dogs we used five injections (Sampaolesi et al. Nature 2006) and in patients we performed 4 or 5 injections (Cossu et al. EMM 2015). At variance with AAV, multiple injections are possible with mesoangioblasts, but our data show that one single injection of xenogeneic DMD, genetically corrected cells was sufficient to detect the protein by WB while this was not the case injecting wt cells. Thus, fewer injections will likely be needed in a clinical setting.

Referee #4 (Comments on Novelty/Model System for Author):

This is an important study and I felt that the paper certainly has merit in terms of novelty and potential translational implications. The main conclusion of the study is that transplantation of myogenic cells corrected by snRNA-based exon skipping can correct distal (non-transduced) nuclei in myofibers of dystrophic mice. Most importantly, this resulted in an unexpected disproportionate beneficial effect both in terms of (i) dystrophin expression levels and (ii) phenotypic correction efficiencies based on a motility assay. This study has important implications by demonstrating that clinically meaningful levels of dystrophin expression could be attained, even in the face of relatively low engraftment efficiencies of gene-modified myogenic cells by virtue of the diffusion of the exon-skipping snRNA towards non-transduced nuclei in the myofibers. Even if exon-skipping or the use of gene-corrected mesoangioblast is not novel in as of itself, the conclusions of the distal disproportionate effects on dystrophin expression and phenotypic corrections are sufficiently novel and clinically relevant. Previous experiments in published papers based on exon-skipping or mesoangioblasts were not designed to address this important question.

Referee #4 (Remarks for Author):

The author adequately addressed my comments. in particular, the inclusion of the data obtained after intra-arterial delivery significantly strengthen the manuscript.

We thank the Referee for the positive evaluation of our work.

22nd Jan 2024

Dear Dr. Galli,

We are pleased to inform you that your manuscript is accepted for publication and is now being sent to our publisher to be included in the next available issue of EMBO Molecular Medicine.
